# Continuous Evolution Pool ⚡:
# Taming Recurring Concept Drift in Online Time Series Forecasting

## Abstract

Recurring concept drift, a type of concept drift in which previously observed data patterns reappear after some time, is one of the most prevalent types of concept drift in time series. As time progresses, concept drift occurs, and previously encountered concepts are forgotten, thereby leading to a decline in the accuracy of online predictions. Existing solutions mainly employ parameter updating techniques to delay forgetting; however, this may result in the loss of some previously learned knowledge while neglecting the exploration of knowledge retention mechanisms. To retain all knowledge and fully utilize it when the concepts recur, we propose the **C**ontinuous **E**volution **P**ool (`CEP`), a pooling mechanism that stores different instances of forecasters for different concepts. Our method first selects the forecaster nearest to the test sample and then learns the features from its neighboring samples—a process we refer to as *retrieval*. If there are insufficient neighboring samples, it indicates that a new concept has emerged, and a new model will evolve from the current nearest sample to the pool to *store* the knowledge of the concept. Simultaneously, the *elimination* mechanism will enable outdated knowledge to be cleared to ensure the prediction effect of the forecasters. Experiments on real-world datasets demonstrate that by retaining distinct conceptual knowledge, `CEP` significantly enhances online forecasting accuracy, reducing the error by over 20%.

**Resources**: https://anonymous.4open.science/r/CEP-42BA

## 1 Introduction

Accurate time series forecasting, a fundamental task in fields such as finance, energy management, traffic prediction, and environmental monitoring, empowers better decision-making, optimized resource allocation, and effective risk management (Jin et al., 2024). However, in practical applications, online time series forecasting frequently encounters a substantial challenge known as *concept drift*. This is particularly critical as modern data often arrives in continuous streams, rendering traditional offline models trained on static historical data, making them quickly outdated. To maintain accuracy in such dynamic environments, there is a crucial need for online forecasting models that can adapt incrementally without the need for frequent and computationally expensive retraining. This refers to the change in the relationship between input variables and their true values over time. In the context of time series, as time elapses, the underlying patterns and behaviors governing the data can shift, making previously learned models less effective. *Recurring concept drift* is a particularly prevalent and intricate form of concept drift in time series data shown in Figure 1(a). It is characterized by the periodic reappearance of certain concepts after a period of absence. For instance, seasonal patterns may cause similar consumption patterns to recur annually or monthly in electricity consumption time series. Nevertheless, as the time series evolves, the model may forget these previously encountered concepts during the non-recurrence periods, leading to a decline in the accuracy of online predictions examplified in Figure 1(b).

Various methods have been devised to tackle concept drift in the realm of online time series forecasting, but each has its own set of limitations. Methods based on Experience Replay (ER) (Chaudhry et al., 2019), such as DER++ (Buzzega et al., 2020), store historical information in a buffer for learning. However, in delay feedback settings, the experience pool quickly becomes outdated as the stored

Figure 1: Concept recurrence is a notable phenomenon in time series data. For instance, in (a), the second concept recurs following the emergence of the fourth concept. During online updates and concept drifts, forecasters often suffer from forgetting previously learned concepts, as in (b). To address this challenge and effectively track recurring concepts, the Continuous Evolution Pool employs an evolving and retrieval strategy, assigning distinct forecasters to manage different concepts.

samples fail to reflect the latest data distribution, and the restricted memory size often leads to the forgetting of once emerged concepts, thus degrading performance when adapting to the evolving data. FSNet (Pham et al., 2023), drawing inspiration from the Complementary Learning Systems theory, tries to balance rapid adaptation to novel changes and the retrieval of past knowledge. However, in scenarios with sample delays (Jhin et al., 2024), it encounters forecasting biases, and makes it harder to accurately capture the patterns in the time series. It also struggles to effectively learn relevant features during intricate concept drifts. OneNet (Zhang et al., 2023) attempts to address concept drift by dynamically updating and integrating models through a reinforcement learning based approach. Nevertheless, its weight adjustment mechanism, which depends on recent prediction errors, fails due to information lag in delay scenarios, and it grapples with handling long-term dependencies in time series data, causing inaccuracies in predicting future trends.

Existing solutions for addressing concept drift in online time series forecasting mainly center around parameter updating techniques. These methods strive to make the model adapt to the changing data distribution by constantly adjusting the model's parameters. Although to some extent, these methods can delay the forgetting of past concepts, they have multiple shortcomings. **First**, parameter updating might lead to the loss of some valuable previously learned knowledge. As the model tries to fit the new data, it could overwrite or distort information related to past concepts. **Second**, these methods generally overlook the exploration of effective knowledge retention mechanisms. Specifically, they do not explicitly store and utilize the knowledge of different concepts in a way that allows for easy retrieval and reuse when those concepts reappear. **Third**, experiments on real-world datasets have demonstrated that neural networks can effectively learn under gradual distribution shifts without requiring elaborate techniques (Read, 2018). In the context of concept abruption (Read, 2018), particularly in scenarios involving recurring concepts, there remains considerable room to improve the performance of existing methods. This is primarily because these methods have difficulty effectively capturing and adapting to sudden changes in data patterns, resulting in suboptimal forecasting accuracy. In light of these limitations, two crucial questions arise:

> **Q1**: *Can we design a mechanism that proactively identifies statistical features of concepts rather than passively waiting for prediction error signals, thereby achieving instant retrieval and zero-loss retention of knowledge?*
>
> **Q2**: *How can we accurately identify distinct concepts and their recurrence, and manage these previously encountered concepts in a resource-efficient manner?*

To address these limitations and effectively manage recurring concept drift in online time series forecasting, we propose the **C**ontinuous **E**volution **P**ool (`CEP`). This is a novel pooling mechanism designed to store multiple forecasters corresponding to different concepts. Its core idea involves partitioning samples from distinct distributions and continuously updating separate models for each evolving concept. When a new test sample arrives, `CEP` selects the forecaster in the pool that is nearest to this sample for prediction, concurrently learning from the features of its neighboring samples. If the neighboring samples are insufficient, this indicates the emergence of a new concept, prompting the evolution of a new model from the closest available sample. This new model is then added to the pool to retain knowledge of the emerging concept. Additionally, `CEP` employs an

elimination mechanism that removes outdated knowledge and filters noisy data, thereby maintaining high forecasting accuracy. The main contributions of this paper are as follows:

1. We identify recurring concept drift as a significant challenge in online time series forecasting, especially under delay feedback scenarios. Existing methods that rely on parameter updating frequently forget previously learned concepts during periods of non-recurrence and lack efficient mechanisms for knowledge retention. This results in reduced prediction accuracy when past concepts reappear.

2. We propose `CEP`, a pooling framework specifically designed to address recurring concept drift under delay feedback. By storing and retrieving forecasters associated with distinct concepts, `CEP` effectively mitigates knowledge loss inherent in traditional methods, enabling the model to leverage previously acquired knowledge more effectively.

3. Through extensive experiments involving multiple real-world datasets and various neural network architectures, we demonstrate that `CEP` substantially enhances online prediction accuracy in scenarios characterized by recurring concept drift. `CEP` consistently outperforms existing methods, particularly in managing complex time series data and delay scenarios.

## 2 RELATED WORK

**Concept Drift.** Unlike conventional time series analysis, where concept drift might be less critical, it is a central concern in online time series forecasting. Suppose the input variables $x$ and their ground truth $y$ follow a distribution $p_t$ over time as Equation 1. For two different time points $t_1$ and $t_2$, the distribution shift can be formally defined by Equation 2 (Read, 2018).

$$(x, y) \sim p_t(x, y) \tag{1}$$

$$p_{t_1}(y|x) \neq p_{t_2}(y|x) \tag{2}$$

Concept drift was formally defined (Tsymbal, 2004), with later work emphasizing the need for continuous adaptation in data streams (Read, 2018). Early models exhibited catastrophic forgetting and lacked rapid learning, hindering real-time adaptation and rarely utilizing deep neural networks. Subsequent work includes the AdaRNN (Du et al., 2021), which struggled with complex relationships, RevIN (Kim et al., 2022) designed for distribution shift, and the exploration into online drift detection (Wan et al., 2024). Despite these advancements, significant challenges remain in online time series forecasting, necessitating continuous research.

**Online Time Series Forecasting.** Unlike traditional methods, online time series forecasting requires continuous learning as new data arrives. Early models (Huszár, 2018; Kirkpatrick et al., 2018) suffered from forgetting past knowledge and lacked rapid adaptation, rarely utilizing deep neural networks. As the field evolved, FSNet (Pham et al., 2023) improved adaptation speed to recent data. To better capture complex relationships and address concept drift, OneNet (Zhang et al., 2023) was introduced, combining cross-variable/cross-time modeling and online ensembling. Despite these advances, challenges persist, requiring ongoing research for improved performance and adaptability.

**Distinctions from Other Drift Adaptation Approaches.** While numerous methods address concept drift in various domains, several prominent approaches are not directly applicable to our specific scenario of recurring concept drift in online time series forecasting. Drift detectors (e.g., ADWIN (Bifet & Gavalda, 2009)) primarily focus on identifying *when* drift occurs but do not provide mechanisms for *how* to retain and manage independent forecasting models for different concepts, particularly in recurring scenarios. Tree-based models (e.g., ARF (Gomes et al., 2017)), while effective for abrupt drift in classification tasks or single step forecasting, typically exhibit weaker performance in capturing long-range temporal dependencies compared to deep learning models, making them unsuitable for our multi-step forecasting requirements. Dynamically expanding continual learning methods (Bifet & Gavalda, 2009; Gomes et al., 2017; Elwell & Polikar, 2011; Kim et al., 2024) are designed for classification tasks, and their mechanisms are difficult to transfer to our regression scenarios. Furthermore, unlike detectors that provide a binary signal of drift, `CEP` directly identifies *which concept the data resembles, enabling targeted retrieval of specialized knowledge rather than triggering a generic adaptation response.* This proactive, concept-centric mechanism is better suited for handling recurring drifts.

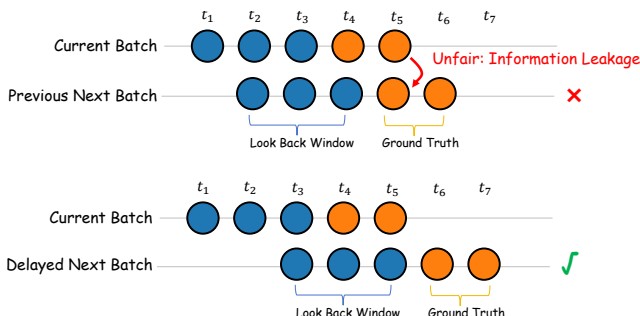

Figure 2: Delayed feedback setting. The blue region represents the input, while the yellow region denotes the corresponding ground truth. In the previous online forecasting setting, the model knows part of the ground truth for the next sample beforehand. This situation is considered *unfair*. In the real world, the model must predict the future values of the ground truth. Therefore, a delay setting is used here.

## 3 METHODOLOGY

In this section, we introduce the proposed `CEP`. It helps partition samples from different distributions, enabling the updating of different models for evolution. We describe how to utilize the dynamic evolution mechanism for the recurring concept scenario. For **Q1**, the evolution mechanism is introduced that assign the different concepts for different forecasters in Section 3.2 **Nearest Evolution**. For **Q2**, some adaptation techniques such as Elimination are introduced to manage the previously learned concepts in Section 3.2 **Forecaster Elimination**. The detailed notations are summarized in Appendix Table 4. A practical guide for choosing hyperparameters can be found in Appendix Table 11.

**Data Format.** A *univariate* time series is represented as $x = (x_1, x_2, \ldots, x_N)$. Consider a look-back window of length $\mathbf{L}$, which is denoted as $\mathbf{x}_i = (x_i, x_{i+1}, \ldots, x_{i+\mathbf{L}-1})$. The corresponding ground truth $\mathbf{y}_i$ with a forecasting horizon of $\mathbf{H}$ is defined as $\mathbf{y}_i = (x_{i+\mathbf{L}}, x_{i+\mathbf{L}+1}, \ldots, x_{i+\mathbf{L}+\mathbf{H}-1})$. Each pair of $\mathbf{x}_i$ and $\mathbf{y}_i$ forms an instance pair expressed as $\mathcal{D}_i = (\mathbf{x}_i, \mathbf{y}_i)$.

**Problem Formulation.** In contrast to traditional time series forecasting, online forecasting is partitioned into two distinct stages: (1) the warm-up stage and (2) the online stage. The dataset was divided into the two stages at a specific ratio. Once the dataset $x$ is split into these two stages, sample pairs are constructed for each of them. The warm-up stage setting is consistent with traditional time series forecasting. In the online stage, the step size of the instance pair is set as forecasting horizon in Figure 2. This setting effectively prevents the forecasting of the ground truth in the overlapping parts, which is in line with the setup of delayed feedback adopted in previous research (Pham et al., 2023; Zhang et al., 2023). The batch size and epoch for the online stage are both set to 1. For *univariate time series without auxiliary information from other channels*, it is necessary to focus on the conceptual changes of the single channel itself. Previous research (Zhang et al., 2023) has shown that time-channel independent online forecasting can reduce interference from too many channels.

### 3.1 WARM-UP STAGE

**Pool Initialization.** We first initialize a pool set $Pl = \{(f(\theta_1), \mathbf{g}_1)\}$ with a single element. Here, $f(\theta_1)$ represents the initialized ancestor forecaster with parameter $\theta_1$, and $\mathbf{g}_1$ is the corresponding gene vector that stores the features of the instances. The gene, serving as a statistical signature for the underlying data distribution, is composed of two parts with a gene ratio $\tau_g$ as defined in Equation 3: the global gene vector $\tilde{\mathbf{g}}$ and the local gene vector $\check{\mathbf{g}}$. The global gene $\tilde{\mathbf{g}}$ is designed to capture and record long-term macroscopic features, which are crucial for forecasting. In contrast, the local gene $\check{\mathbf{g}}$ is responsible for documenting short-term features that are strongly correlated with the current time. During the warm-up stage, which has the same task as typical time series forecasting, `CEP` employs the ancestor forecaster to make forecasts and accumulate knowledge for the online stage with recurring concepts.

$$\mathbf{g} = \tau_g * \check{\mathbf{g}} + (1 - \tau_g) * \tilde{\mathbf{g}} \tag{3}$$

Figure 3: Illustration of the proposed `CEP` mechanism: (a) **Adaptive Updating**: Continuously update the position of the evolved forecaster in the gene space as new instances arrive. (b) **Nearest Evolution**: If the instance surpasses the forecaster's evolution threshold, evolve and replicate the closest forecaster at the instance. (c) **Optimizer Adjustment**: Adjust the learning rate for the shifted concept ■ to ensure accurate adaptation. (d) **Forecaster Elimination**: The forecaster $f_4$ associated with the rarely-occurring noise concept ■ may be removed due to prolonged inactivity. (e) **Nearest Retrieval**: Identify the nearest forecaster in the gene space when an input instance is encountered.

**Adaptive Updating.** The forecaster genes are adaptively updated as instances arrive, stabilizing over time (Figure 3a). Each instance $\mathbf{x}$ is characterized by a gene $\mathbf{g}_x = (\mu(\mathbf{x}[-S:]), \sigma(\mathbf{x}[-S:]))$ (Equation 4), where $S$ limits the look-back window scope. Local genes are updated with ratio $\tau_l$ for short-term adaptation (Equation 5), while global genes employ statistical formulas for overall trend estimation (Equation 6). The online update mechanism is derived in Appendix Section B.4.

$$\mathbf{g}_x = \text{Gene}(x) = (\mu(\mathbf{x}[-S:]), \sigma(\mathbf{x}[-S:])) \tag{4}$$

$$\check{\mathbf{g}} \leftarrow \tau_l * \mathbf{g}_x + (1 - \tau_l) * \check{\mathbf{g}} \tag{5}$$

$$\tilde{\mathbf{g}} \leftarrow \left( \frac{n \cdot \tilde{\mathbf{g}}_\mu + \mathbf{g}_{x,\mu}}{n+1}, \sqrt{\frac{n}{n+1}\tilde{\mathbf{g}}_\sigma^2 + \frac{n}{(n+1)^2}(\tilde{\mathbf{g}}_\mu - \mathbf{g}_{x,\mu})^2} \right) \tag{6}$$

Additionally, *adaptive updating is not only limited to the warm-up stage but also applies to the online stage*. In the case of continuous concept drift shown in Section 4.2, the forecaster can adapt naively. This should not lead to the evolution of more forecasters, causing the samples to be dispersed, and making it difficult for all forecasters to be well-trained. The updating behaviour can enable the forecaster to follow the slight and continuous changes in the data.

## 3.2 ONLINE STAGE

**Nearest Retrieval.** For a new instance $\mathbf{x}$ with gene $\mathbf{g}_x$, the closest forecaster is retrieved using Euclidean distance (Equation 7, Equation 8). This hard assignment strategy ensures conceptual purity by dedicating each forecaster to a specific data distribution, with theoretical justification in Appendix Section B.3.

$$d(\mathbf{g}_x, \mathbf{g}) = \|\mathbf{g}_x - \mathbf{g}\|_2 \tag{7}$$

$$(f(\theta_N), \mathbf{g}_N) = \underset{(f(\theta_i), \mathbf{g}_i) \in Pl}{\text{argmin}} d(\mathbf{g}_x, \mathbf{g}_i) \tag{8}$$

**Nearest Evolution.** Given that the instances in online time series forecasting arrive incrementally, the features of different concepts may exhibit instability during the initial stages. To ensure the stability of the gene, each forecaster in the pool $Pl$ is assigned a safety period $\tau_{safe}$. During the online phase, if the number of forecasting instances completed by $f(\theta_N)$ is fewer than the safety period, no splitting operation is executed. Conversely, when the number of completed forecasting instances by $f(\theta_N)$ reaches or exceeds the safety period, the splitting threshold $\tau_s$ is determined. Once this threshold is exceeded, it implies that a distribution shift has occurred in the current instance.

The threshold $\tau_s$ is established to detect the recurring concepts. Specifically, the threshold criteria $\tau_\mu$ and $\tau_\sigma$ for the mean and standard deviation which is an efficient implemention of $\tau_s$ are defined in Equation 9 and Equation 10, respectively. If the threshold is surpassed, the evolution process, as described in Equation 11, is initiated. Upon evolution, a split forecaster is added to the evolution pool $Pl$. The newly created forecaster inherits the parameters of the nearest forecaster $f(\theta_N)$, and simultaneously, its corresponding gene inherits the gene $g_x$ of the input instance, as shown in Figure 3(b). *This choice leverages transfer learning; by initializing from the closest existing model, the new forecaster starts with a strong inductive bias, significantly accelerating convergence on the new but related concept compared to initializing from scratch.*

$$|\mathbf{g}_{x,\mu} - \mathbf{g}_{N,\mu}| > \tau_\mu \cdot \mathbf{g}_{N,\sigma} \tag{9}$$

$$\mathbf{g}_{x,\sigma} > \tau_\sigma * \mathbf{g}_{N,\sigma} \tag{10}$$

$$Pl \leftarrow Pl \cup \{(f(\theta_N), \mathbf{g}_x)\} \tag{11}$$

The evolution thresholds implement a form of online hypothesis testing where deviations beyond $\tau_\mu$ standard deviations from the concept mean indicate a high probability of concept drift. Detailed statistical justification is provided in Appendix Section B.2. Subsequently, the split forecaster will be tasked with forecasting the instance $\mathbf{x}$ and updating its parameters. *It is important to note that in online forecasting, the normalization layer typically maintains the current concept. For CEP, the evolution mechanism serves a similar purpose.* On one hand, the evolution mechanism ensures that the samples learned by the forecasters in the pool belong to individual concepts. On the other hand, CEP does not focus on the internal design of the forecaster, aiming to enhance generalization and accommodate a wider range of forecasters.

**Forecaster Elimination.** Forecasters idle for extended periods are removed to mitigate errors from time series fluctuations and conserve storage (Figure 3d). A threshold $\tau_e$ triggers elimination when $n_{wait} > \tau_e \cdot n_{pred}$ (Equation 12), where $n_{wait}$ is idle time which refers to the number of time points that have passed since the last prediction. and $n_{pred}$ is total predictions. This mechanism promptly removes erroneously evolved forecasters and prevents noise corruption.

$$n_{wait} > \tau_e * n_{pred} \tag{12}$$

**Optimizer Adjustment.** For abrupt concept shifts (Figure 3c), the learning rate is adjusted as $lr = \tau_{lr} \cdot lr_{raw}$ initially, then gradually restored via exponential decay (Equation 13), where $t_{lr}$ is inspiration time. This balances rapid adaptation with stable convergence.

$$lr \leftarrow max(lr_{raw}, \tau_{lr}^{-\frac{1}{t_{lr}}} * lr) \tag{13}$$

**Gradient Abandonment.** If the ground truth gene $\mathbf{g}_y$ triggers a split (per Equation 9, Equation 10), parameter updating is skipped to prevent concept contamination from anticipated distribution shifts, preserving forecaster specialization.

### 3.3 HYPERPARAMETER CONFIGURATION

The thresholds in our user-driven algorithm are configurable, allowing performance to be tuned; for example, the mean threshold $\tau_\mu$ in Equation 9 influences splitting frequency, with lower values promoting more aggressive divisions similar to temperature. This study employs empirically determined parameters. Specifically, $\tau_\mu$ is set to 3, aligning our splitting criterion with the statistical convention of a $3\sigma$ rule, commonly used in Z-test to detect significant deviations from an established mean based on its standard deviation. This facilitates a statistically grounded approach to identifying shifts warranting new forecaster evolution. At the same time, we conducted sensitivity tests for different thresholds to provide a reference for the performance of the CEP in Appendix Section C.6.

## 4 EXPERIMENTS

### 4.1 EXPERIMENT SETTING

**Datasets.** During the experiments, we employed a diverse array of datasets to comprehensively evaluate the time series forecasting model. These datasets encompass: the ECL dataset (Trindade, 2015), the ETT dataset (comprising 4 distinct subsets) Zhou et al. (2021), the Exchange dataset (Wu et al., 2021), the Traffic dataset (Wu et al., 2021), and the WTH dataset (Wu et al., 2021).

**Baselines.** In our experiments, we evaluated multiple baselines across the domains of continual learning, time series forecasting, and online learning. The Experience Replay (ER) method (Chaudhry et al., 2019) stores historical data in a buffer and interleaves it with new samples during the learning process. DER++ (Buzzega et al., 2020) incorporates a knowledge distillation strategy to enhance performance. FSNet (Pham et al., 2023) utilizes a fast-slow learning mechanism to capture both short-term changes and long-term patterns in online time series forecasting. OneNet (Zhang et al., 2023) improves time series forecasting under concept drift through online ensembling, which combines multiple models to adapt to evolving data distributions and boost performance. For time series forecasting, we screened models based on various backbones, including: DLinear (Zeng et al., 2023), TimesNet (Wu et al., 2023), PatchTST (Nie et al., 2023), SegRNN (Lin et al., 2023), iTransformer (Liu et al., 2024), and TimeMixer (Wang et al., 2024). In all benchmarks, the length of the look-back window was set to $\mathbf{L} = 60$, and the forecast horizon was varied as $\mathbf{H} \in \{30, 60\}$. The metric used for evaluation were Mean Squared Error (MSE).

### 4.2 MAIN RESULTS

#### 4.2.1 COMPARISON WITH NAIVE FORECASTING MODELS

Experiment results are presented in Table 1. The proposed CEP generally reduces the MSE in forecasting across diverse datasets and model architectures. This reduction in MSE indicates that CEP effectively captures and utilizes the features of different concepts, thereby enhancing the performance of online time series forecasting in scenarios with recurring concept drift. CEP's effectiveness is not limited to a single architecture or dataset, demonstrating its broad applicability. Notably, the TCN model (Pham et al., 2023) experiences a particularly significant improvement in univariate online forecasting scenarios, with the MSE reduction exceeding 20%. This substantial improvement can be attributed to the unique characteristics of TCN, which leverages both time and variable dimensions in its design. In contrast, models like PatchTST (Nie et al., 2023) rely solely on the time dimension, potentially limiting their ability to adapt to complex concept drifts. However, CEP may not yield improvements in specific cases, such as the Traffic dataset. We attribute this to the nature of its concept drifts. The Traffic dataset exhibits frequent, low-magnitude fluctuations rather than the distinct, recurring concepts CEP is designed to capture. Consequently, our statistically-grounded evolution threshold ($\tau_\mu = 3$) is seldom triggered, causing CEP to default to the behavior of the naive model. This highlights an applicable boundary of our method and suggests that a lower evolution threshold may be required for datasets with more subtle drift patterns.

#### 4.2.2 COMPARISON WITH ONLINE FORECASTING MODELS

Experiment results are presented in Table 2. ER, DER++, FSNet, and OneNet did not yield satisfactory performance in delayed scenarios. Experience replay methods (Chaudhry et al., 2019; Buzzega et al., 2020) reuse instances stored in memory to update the model. However, as time progresses, due to the limited memory capacity, the samples in memory may fail to retain the concepts that emerged earlier because of concept drift. FSNet (Pham et al., 2023) is prone to forecasting biases, an inevitable consequence of the delayed arrival of samples. OneNet (Zhang et al., 2023) takes short-term information into account to adjust model parameters, thus optimizing forecasting results to some degree in delayed scenarios. To preserve the knowledge of different concepts comprehensively, the proposed CEP employs a pooling mechanism to assign different concepts to distinct models. In delay-feedback scenarios, nearly all the improvements achieved by CEP significantly outperform those of previous time series methods. This suggests that in delay scenarios, especially those with recurring concepts, previous online methods do not perform as effectively as anticipated and may even be inferior to the naive model shown in Table 2. In contrast, CEP can effectively manage recurring concepts.

Table 1: The averaged MSE errors of different forecasters when using `CEP` are presented. Enhanced and reduced outcomes are marked with 🟥 and 🟩 respectively. The **best** and second-best performances are highlighted. The full results are presented in Appendix Table 8.

| Data | TimeMixer | +CEP⚖ | iTransformer | +CEP⚖ | PatchTST | +CEP⚖ | DLinear | +CEP⚖ | SegRNN | +CEP⚖ | TimesNet | +CEP⚖ | TCN | +CEP⚖ |
|------|-----------|-------|--------------|-------|----------|-------|---------|-------|--------|-------|----------|-------|-----|-------|
| ECL | 0.290 | 0.271 | 0.275 | 0.256 | 0.310 | 0.297 | 0.307 | 0.299 | 0.278 | 0.276 | 0.305 | 0.289 | 0.344 | 0.309 |
| | - | -6.62% | - | -6.98% | - | -4.29% | - | -2.60% | - | -0.79% | - | -5.34% | - | -10.16% |
| ETTh1 | 0.335 | 0.322 | 0.326 | 0.323 | 0.332 | 0.327 | 0.337 | 0.308 | 0.303 | 0.301 | 0.351 | 0.341 | 0.463 | 0.459 |
| | - | -3.91% | - | -0.80% | - | -1.75% | - | -8.70% | - | -0.76% | - | -2.96% | - | -0.76% |
| ETTh2 | 2.630 | 2.508 | 2.566 | 2.542 | 2.594 | 2.458 | 2.718 | 2.630 | 2.628 | 2.584 | 2.676 | 2.560 | 3.166 | 2.804 |
| | - | -4.65% | - | -0.93% | - | -5.23% | - | -3.22% | - | -1.66% | - | -4.37% | - | -11.44% |
| ETTm1 | 0.755 | 0.737 | 0.760 | 0.740 | 0.759 | 0.721 | 0.849 | 0.752 | 0.719 | 0.716 | 0.837 | 0.793 | 0.981 | 0.801 |
| | - | -2.39% | - | -2.64% | - | -5.00% | - | -11.53% | - | -0.32% | - | -5.21% | - | -18.40% |
| ETTm2 | 0.237 | 0.235 | 0.240 | 0.238 | 0.237 | 0.236 | 0.235 | 0.227 | 0.232 | 0.228 | 0.258 | 0.255 | 0.271 | 0.264 |
| | - | -0.76% | - | -0.79% | - | -0.55% | - | -3.36% | - | -1.85% | - | -1.05% | - | -2.48% |
| Exchange | 0.356 | 0.342 | 0.385 | 0.367 | 0.371 | 0.350 | 0.408 | 0.377 | 0.295 | 0.294 | 0.422 | 0.394 | 0.542 | 0.412 |
| | - | -4.02% | - | -4.73% | - | -5.61% | - | -7.44% | - | -0.51% | - | -6.59% | - | -23.90% |
| Traffic | 0.478 | 0.478 | 0.486 | 0.486 | 0.586 | 0.586 | 0.628 | 0.628 | 0.812 | 0.812 | 0.544 | 0.544 | 0.691 | 0.691 |
| | - | 0.00% | - | 0.00% | - | 0.00% | - | 0.00% | - | 0.00% | - | 0.00% | - | 0.00% |
| WTH | 0.356 | 0.355 | 0.353 | 0.352 | 0.354 | 0.353 | 0.342 | 0.340 | 0.341 | 0.342 | 0.364 | 0.364 | 0.376 | 0.382 |
| | - | -0.31% | - | -0.23% | - | -0.23% | - | -0.35% | - | 0.21% | - | -0.25% | - | 1.44% |

Table 2: The average MSE error of previous online forecasting methods. The previous methods all integrated TCN according to the original settings, and `CEP`'s forecaster is also TCN. Enhanced and reduced outcomes are marked with 🟥 and 🟩 respectively, the percentages represents the difference compared to TCN. The **best** and second-best performances are highlighted. Full results are presented in Appendix Table 9.

| Data | ECL | ETTh1 | ETTh2 | ETTm1 | ETTm2 | Exchange | Traffic | WTH |
|------|-----|-------|-------|-------|-------|----------|---------|-----|
| TCN | 0.344 | 0.463 | 3.166 | 0.981 | 0.271 | 0.542 | 0.691 | **0.376** |
| ER | 0.624 | 0.454 | 3.329 | 1.382 | 0.322 | 2.909 | 1.110 | 0.558 |
| | 81.16% | -1.88% | 5.16% | 40.81% | 19.06% | 437.19% | 60.60% | 48.29% |
| DER++ | 0.580 | **0.407** | 3.073 | 1.213 | 0.304 | 2.515 | 1.076 | 0.529 |
| | 68.26% | -12.12% | -2.94% | 23.61% | 12.38% | 364.47% | 55.67% | 40.69% |
| FSNet | 0.843 | 0.470 | 4.072 | 1.562 | 0.347 | 3.486 | 1.466 | 0.743 |
| | 144.72% | 1.51% | 28.61% | 59.17% | 28.15% | 543.77% | 112.05% | 97.53% |
| OneNet | 0.425 | 0.410 | 3.402 | 1.093 | 0.285 | 1.130 | **0.673** | 0.414 |
| | 23.49% | -11.47% | 7.47% | 11.39% | 5.10% | 108.72% | -2.65% | 10.12% |
| CEP⚖ | 0.309 | 0.459 | **2.804** | **0.801** | **0.264** | **0.412** | 0.691 | 0.382 |
| | -10.16% | -0.76% | -11.44% | -18.40% | -2.48% | -23.90% | 0.00% | 1.44% |

## 4.3 VISUALIZATION

Figures 4 and 5 illustrate the forecasting results and MSE of both the naive model and the `CEP`. When the concept recurs (instances 2-14), the naive model exhibits a significant performance degradation, as evidenced by the surge in MSE. In contrast, `CEP` maintains a stable and low MSE, demonstrating its effective adaptation to the recurring concept. Figure 6 provides insight into the internal mechanism of `CEP`. The different colors represent distinct forecasters within the pool. The visualization confirms that `CEP` promptly activates a specialized forecaster (Forecaster 1) that has learned the features of the recurring concept. This targeted activation is the reason for its superior performance, as it avoids catastrophic forgetting and leverages stored knowledge.

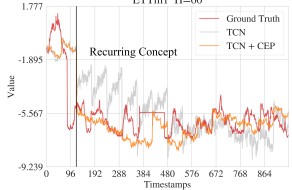 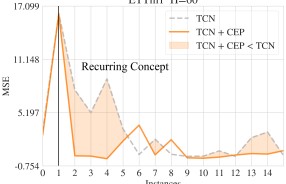 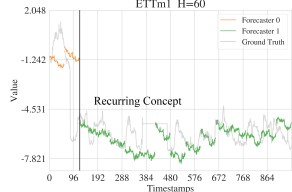

Figure 4: Visualization of forecasting results

Figure 5: Visualization of Metric

Figure 6: Visualization of different forecaster results in `CEP`

## 4.4 COMPUTATION COMPLEXITY

Figure 7 presents an empirical analysis of the computational complexity, comparing `CEP` against naive TCN and other online methods. **In terms of time complexity**, `CEP` is highly efficient. As it activates only a single specialized forecaster per instance, its inference time is identical to that of the base forecaster model. **Regarding space complexity**, the *Forecaster Elimination* mechanism is crucial for preventing the unbounded growth of the forecaster pool. By dynamically pruning forecasters

associated with outdated or transient concepts, this mechanism ensures the memory footprint remains practical and bounded. A more formal analysis is provided in Appendix D.

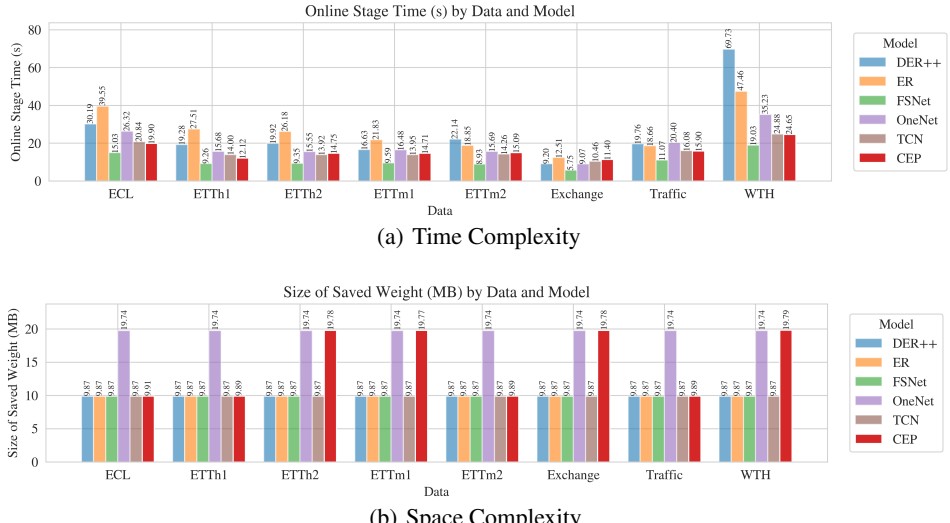

(a) Time Complexity

(b) Space Complexity

Figure 7: Computation Complexity of naive TCN, online methods and `CEP`

## 4.5 MODEL ANALYSIS

To validate the efficacy of each component in `CEP`, we conducted extensive ablation studies, with the results presented in Table 3. A key finding is that the evolutionary mechanism is indispensable; for instance, disabling Evolution for the TCN forecaster leads to a substantial increase in prediction error, demonstrating that our evolutionary framework is crucial for performance. The complete `CEP`, referred to as CEP, yielded the best results, significantly outperforming all ablated versions. Interestingly, the second-best performance was achieved without the Elimination mechanism. This highlights the dual role of elimination: it not only maintains a manageable pool size but also discards outdated knowledge, ensuring the model's continuous adaptation. Comprehensive ablation results and other experimental details, including parameter sensitivity, are provided in Appendix C.

Table 3: The MSE error of the ablation experiments with forecaster TCN. The results are the mean values of different backbone forecasters. The **best** and second-best performances are highlighted. Full results are presented in Appendix Table 10.

| Dataset | | ECL | ETTh1 | ETTh2 | ETTm1 | ETTm2 | Exchange | Traffic | WTH | **AVG** |
|---|---|---|---|---|---|---|---|---|---|---|
| CEP 🏆 | 🏅 | **0.314** | 0.438 | 2.799 | **0.801** | 0.256 | 0.404 | 0.710 | 0.379 | **0.762** |
| Naive TCN (*w/o* Evolution) | 👎 | 0.347 | **0.436** | 3.106 | 0.991 | 0.261 | 0.524 | 0.710 | 0.375 | 0.844 |
| *w/o* Fast Gene | | 0.317 | 0.451 | 2.904 | **0.801** | 0.257 | 0.524 | 0.710 | 0.375 | 0.792 |
| *w/o* Slow Gene | | 0.321 | 0.438 | 2.872 | 0.865 | 0.256 | **0.401** | 0.710 | **0.367** | 0.779 |
| *w/o* Elimination | 🏅 | 0.335 | 0.438 | **2.797** | **0.801** | **0.254** | 0.406 | 0.710 | 0.379 | 0.765 |

## 5 CONCLUSION

In this paper, we address the critical challenge of recurring concept drift in online time series forecasting, where models often suffer from catastrophic forgetting. We introduce `CEP`, an evolutionary framework that maintains a diverse pool of specialized forecasters to preserve learned concepts. By evolving new forecasters from the most relevant existing ones and eliminating outdated knowledge, `CEP` effectively adapts to concept shifts. Our experiments demonstrate that `CEP` significantly enhances long-horizon forecasting performance, overcoming the limitations of current online methods that struggle with knowledge retention. Future work will explore richer gene representations, potentially incorporating higher-order moments or spectral features to address more complex drifts, and extend this framework to handle asynchronous shifts in multivariate time series.

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

## A  NOTATIONS

In this section, we provide a summary of the notation used in the main paper, particularly in the method section, for quick reference. Table 4 lists the key symbols and their corresponding descriptions.

Table 4: Summary of important notations.

| Notation | Description | Hyperparameter |
|:---:|:---:|:---:|
| $p$ | Probability distribution of time series | |
| $x$ | Original time series | |
| $\mathbf{x}_i$ | Input window | |
| $\mathbf{y}_i$ | Ground truth window | |
| $\mathbf{L}$ | Input window length | |
| $\mathbf{H}$ | Forecasting horizon | |
| $\mathcal{D}_i$ | Input and ground truth window pair: Data pair | |
| $\mathcal{D}$ | Data pairs | |
| $\widetilde{\mathcal{D}}_i$ | Data pairs of concept $i$ | |
| $Pl$ | Forecaster Pool | |
| $f(\cdot)$ | Forecaster | |
| $\theta_i$ | Forecaster parameter of concept $i$ | |
| $\mathbf{g}_i$ | Gene of concept $i$ | ✗ |
| $\check{\mathbf{g}}$ | Local gene | |
| $\widetilde{\mathbf{g}}$ | Global gene | |
| $\mathbf{g}_x$ | Sample gene | |
| $\mu$ | Mean | |
| $\sigma$ | Standard deviation | |
| $S$ | Scope of $\mu$ and $\sigma$ | |
| $n$ | Number of historical samples | |
| $d(\cdot, \cdot)$ | Distance of gene | |
| $N$ | Nearest neighbor | |
| $n_{wait}$ | Number of wait time of forecaster | |
| $n_{pred}$ | Number of prediction of forecaster | |
| $lr$ | Learning rate | |
| $\tau_g$ | Global gene ratio | |
| $\tau_l$ | Fast gene ratio | |
| $\tau_s$ | Splitting threshold | |
| $\tau_\mu$ | Mean threshold for splitting | ✓ |
| $\tau_\sigma$ | Standard deviation threshold for splitting | |
| $\tau_{safe}$ | Intialization time of forecaster | |
| $\tau_e$ | Elimination threshold of forecaster | |
| $\tau_{lr}$ | Adjustment ratio of learning rate | |

## B  TECHNICAL DETAILS

### B.1  ALGORITHM MOTIVATION

The core principle of the Continuous Evolution Pool (CEP) is to partition the online data stream into distinct concepts and dedicate a specialized forecaster to each. This approach directly tackles catastrophic forgetting by isolating the learning process for each concept, preventing knowledge from one from being overwritten by another. It contrasts with standard online learning, where a single model continuously adapts, and with Experience Replay (Chaudhry et al., 2019), where stored samples may become outdated or fail to represent all past concepts.

Conceptually, CEP operationalizes the ideal of partitioning a dataset $\mathcal{D}$ into subsets $\widetilde{\mathcal{D}}_i$, where each subset contains instances from a single data distribution $p_i$. In an online setting, this partitioning must be performed dynamically without storing past instances due to privacy or memory constraints. CEP achieves this by using a statistical gene to identify the concept of each incoming instance $\mathcal{D}_i$. It then assigns the instance to the forecaster with the most similar gene, ensuring that each forecaster

is trained only on data from its designated concept. This process effectively handles the challenges of online learning, such as imbalanced data and the non-sequential arrival of concepts, by promptly selecting an appropriate, pre-specialized model. The complete workflow is presented in Algorithm 1.

---

**Algorithm 1:** Continuous Evolution Pool

**Initialize**

  Dataset $x$ ;

  Forecaster $f$ ;

  $\mathcal{D}_{warm}, \mathcal{D}_{online} \leftarrow \text{split}(x)$ ;                     ▷ Problem Formulation

  Initialize $Pl = \{(f(\theta_1), g_1)\}$ ;

**Stage** *Warm-up*

  **for** $\mathcal{D}_i \in \mathcal{D}_{warm}$ ;                          ▷ $\mathcal{D}_i = (\mathbf{x}_i, \mathbf{y}_i)$

  **do**

    $g_x = \text{Gene}(\mathbf{x}_i)$ ;                   ▷ Eq.Equation 4

    $\mathcal{L} = \text{Loss}(f(\mathbf{x}_i, \theta_1), \mathbf{y}_i)$ ;

    $\theta_1 \leftarrow \theta_1 - lr * \nabla\mathcal{L}$ ;

    $g_1 \leftarrow \text{Update}(g_1, g_x)$ ;         ▷ Update gene, Eq.Equation 3,
     Eq.Equation 5, Eq.Equation 6

**Stage** *Online*

  **for** $\mathcal{D}_i \in \mathcal{D}_{online}$ ;                       ▷ $\mathcal{D}_i = (\mathbf{x}_i, \mathbf{y}_i)$

  **do**

    $g_x = \text{Gene}(\mathbf{x}_i)$ ;                   ▷ Eq.Equation 4

    $(f(\theta_n), g_n) = \text{Nearest}(Pl, g_x)$ ;      ▷ Eq.Equation 7, Eq.Equation 8

    **if** *Evolution*$(g_x, g_n)$ ;        ▷ Evolve or retrieve, Eq.Equation 9,
     Eq.Equation 10

    **then**

      $\theta_c = \text{Copy}(\theta_n)$ ;               ▷ Evolve

      $Pl \leftarrow Pl \cup (f(\theta_c), g_x)$ ;

      $lr \leftarrow \text{Adjust}(lr)$ ;        ▷ Adjust optimizer, Eq.Equation 13

    **else**

      $\theta_c = \theta_n$ ;                     ▷ Retrieve

    $\hat{\mathbf{y}}_i = f(\mathbf{x}_i, \theta_c)$ ;

    $g_y = \text{Gene}(\mathbf{y}_i)$ ;                ▷ Eq.Equation 4

    **if** *not Evolution*$(g_y, g_n)$ ;      ▷ Give up the polluted gradient,
     Eq.Equation 9, Eq.Equation 10

    **then**

      $\mathcal{L} = \text{Loss}(\hat{\mathbf{y}}_i, \mathbf{y}_i)$ ;

      $\theta_c \leftarrow \theta_c - lr * \nabla\mathcal{L}$ ;

      $g_c \leftarrow \text{Update}(g_c, g_x)$ ;      ▷ Update gene, Eq.Equation 3,
      Eq.Equation 5, Eq.Equation 6

    $Pl \leftarrow \text{Eliminate}(Pl)$ ;      ▷ Eliminate ineffective forecasters,
     Eq.Equation 12

---

### B.2 JUSTIFICATION FOR GENE-BASED CONCEPT REPRESENTATION

A fundamental question in the design of `CEP` is why the mean and standard deviation of the input window serve as an effective gene for identifying concepts. This choice is not arbitrary; it is grounded in the statistical definition of concept drift and offers significant advantages in speed and directness over traditional error-driven adaptation methods.

**Concept Drift as a Distributional Shift.** Concept drift, in its most general form, is a change in the underlying data-generating distribution over time. Let's consider the distribution of the input window, $p_t(\mathbf{x})$. A drift from a concept $C_A$ at time $t_1$ to a concept $C_B$ at time $t_2$ implies that the distributions

are different:

$$p_{t_1}(\mathbf{x}) \neq p_{t_2}(\mathbf{x}) \tag{14}$$

This change in the probability distribution will manifest as a change in its statistical properties, most notably its moments. The first two moments, the mean and variance, are defined as:

$$\mathbb{E}_t[\mathbf{x}] = \int \mathbf{x} \cdot p_t(\mathbf{x})d\mathbf{x} \tag{15}$$

$$\mathrm{Var}_t(\mathbf{x}) = \mathbb{E}_t[(\mathbf{x} - \mathbb{E}_t[\mathbf{x}])^2] = \int (\mathbf{x} - \mathbb{E}_t[\mathbf{x}])^2 \cdot p_t(\mathbf{x})d\mathbf{x} \tag{16}$$

If $p_{t_1}(\mathbf{x}) \neq p_{t_2}(\mathbf{x})$, it is highly likely that their means and/or variances will also differ, i.e., $\mathbb{E}_{t_1}[\mathbf{x}] \neq \mathbb{E}_{t_2}[\mathbf{x}]$ or $\mathrm{Var}_{t_1}(\mathbf{x}) \neq \mathrm{Var}_{t_2}(\mathbf{x})$.

**Sample Statistics as Unbiased Estimators.** In a real-world online setting, we do not have access to the true distribution $p_t(\mathbf{x})$. Instead, we work with a finite sample—the look-back window $\mathbf{x}_i$. The instance gene $\mathbf{g}_x = (\mu(\mathbf{x}_i), \sigma(\mathbf{x}_i))$ computed by `CEP` uses the sample mean and sample standard deviation. These are well-known, efficient, and unbiased estimators of the true mean and standard deviation of the underlying distribution from which the sample was drawn.

Therefore, a statistically significant deviation between the gene of a new instance, $\mathbf{g}_{x,i}$, and the established gene of the nearest forecaster, $\mathbf{g}_N$, strongly suggests that the instance $\mathbf{x}_i$ was drawn from a different distribution than the historical samples that formed the concept of forecaster $f_N$.

**Connection to Statistical Hypothesis Testing.** The evolution criteria in Equation 9 and Equation 10 can be interpreted as a form of simplified, online statistical hypothesis testing.

- **Null Hypothesis ($H_0$):** The new instance $\mathbf{x}_i$ belongs to the same concept as the nearest forecaster $f_N$. In other words, $\mathbf{x}_i \sim p_N(\mathbf{x})$.

- **Alternative Hypothesis ($H_1$):** The new instance $\mathbf{x}_i$ belongs to a different concept.

The condition $|\mathbf{g}_{x,\mu} - \mathbf{g}_{N,\mu}| \leq \tau_\mu * \mathbf{g}_{N,\sigma}$ is analogous to a Z-test. It assesses whether the new sample's mean falls within a confidence interval (defined by $\tau_\mu$ standard deviations) around the concept's established mean. If it falls outside this interval, we reject the null hypothesis and conclude that a concept drift has likely occurred, triggering an evolution. Using the forecaster's standard deviation $\mathbf{g}_{N,\sigma}$ as the scaling factor makes the test adaptive to the natural volatility of each concept.

**Advantages over Error-based Detection.** This gene-based approach offers two primary advantages over methods that rely on monitoring prediction error:

1. **Speed and Proactiveness.** Changes in the input data distribution are a leading indicator of concept drift. By monitoring these statistics directly, `CEP` can detect a drift as soon as it occurs. In contrast, error-based methods are reactive; they must wait for the model to make several poor predictions on the new concept's data before the increase in error becomes significant. This is especially critical in settings with delayed feedback, where the error signal might arrive too late for effective adaptation.

2. **Directness and Robustness.** Prediction error is an indirect signal of drift. High error could be caused by a true drift, but it could also be due to inherent data noise or a momentary anomaly. The statistical properties of the input data, however, provide a more direct and robust signal of a fundamental change in the underlying data-generating process. This makes the detection mechanism less susceptible to noise and more focused on genuine, persistent shifts.

In summary, by defining concepts via the mean and variance of the input data, `CEP` grounds its core mechanism in fundamental statistical principles. This allows it to detect recurring and novel concepts in a manner that is proactive, direct, and computationally efficient, making it particularly well-suited for the challenges of online time series forecasting.

## B.3 RATIONALE FOR GENE-BASED HARD ASSIGNMENT

While many online learning systems rely on soft-weighting or ensembling methods (e.g., Hedge (Krichene et al., 2015) or OneNet (Zhang et al., 2023)), `CEP` adopts a seemingly simpler hard assignment strategy based on statistical genes. This design choice is deliberate and offers distinct advantages in the context of recurring and often abrupt concept drifts typical of time series data. Here, we justify this approach by highlighting the limitations of soft methods and grounding our gene-based selection in statistical principles.

**The Drawbacks of Soft-Weighting in Abrupt Drifts.** Soft-weighting methods assign a weight to every forecaster in a pool and combine their predictions, typically through a weighted average. While effective for gradual drifts, this approach exhibits two key weaknesses in the face of abrupt changes:

- **Prediction Inertia:** When a concept changes suddenly, the majority of forecasters in the pool become instantly incorrect. However, their historical performance grants them non-zero weights, and their erroneous predictions continue to pollute the final ensembled output. This creates a period of high error while the algorithm slowly down-weights the now-obsolete forecasters, a phenomenon we term prediction inertia.

- **Slow Specialization:** The learning signal (loss) is distributed across all models in the ensemble. This can slow down the specialization of the single, correct forecaster for the new concept, as it does not receive the full, undiluted learning signal from the new data instances.

**Advantages of Hard Assignment via Statistical Genes.** `CEP`'s hard assignment mechanism mitigates these issues by being decisive. It uses the instance's gene not to weigh experts, but to directly select the *single most relevant* expert. The choice of the mean ($\mu$) and standard deviation ($\sigma$) as the gene is rooted in sampling theory and the nature of concept drift itself.

A concept drift, by definition, is a change in the data-generating distribution $p_t(\mathbf{x})$. The most fundamental descriptors of any statistical distribution are its moments. The mean and variance represent the first and second moments, respectively, capturing the central tendency and dispersion of the data. From the perspective of sampling theory, the sample mean ($\hat{\mu}_{\mathbf{x}}$) and sample standard deviation ($\hat{\sigma}_{\mathbf{x}}$) of a look-back window $\mathbf{x}$ are efficient estimators of the true moments of the underlying distribution from which the sample was drawn. Therefore, the gene $\mathbf{g}_{\mathbf{x}} = (\hat{\mu}_{\mathbf{x}}, \hat{\sigma}_{\mathbf{x}})$ serves as a robust, low-dimensional proxy for the current concept's underlying distribution.

**Mathematical Justification.** The effectiveness of using Euclidean distance in the gene space can be intuitively justified by framing the retrieval step as a maximum likelihood estimation problem. Assume that each concept $C_k$ can be modeled by a stable underlying distribution, which for simplicity we can approximate with a normal distribution $\mathcal{N}(\mu_k, \sigma_k^2)$. The gene of the corresponding forecaster, $\mathbf{g}_k$, is a stable estimate of $(\mu_k, \sigma_k)$.

Given a new instance window $\mathbf{x}$, our goal is to determine which concept $C_k$ was most likely to have generated it. According to the principle of maximum likelihood, we should choose the concept $k$ that maximizes the probability $p(\mathbf{x}|C_k)$. Assuming the data points within the window are approximately independent and identically distributed, the log-likelihood is:

$$\log p(\mathbf{x}|C_k) = \log \prod_{i=1}^{L} \frac{1}{\sqrt{2\pi\sigma_k^2}} \exp\left(-\frac{(x_i - \mu_k)^2}{2\sigma_k^2}\right) = -\frac{L}{2}\log(2\pi\sigma_k^2) - \frac{1}{2\sigma_k^2}\sum_{i=1}^{L}(x_i - \mu_k)^2 \quad (17)$$

By substituting the sample statistics $\hat{\mu}_{\mathbf{x}} = \frac{1}{L}\sum x_i$ and $\hat{\sigma}_{\mathbf{x}}^2 = \frac{1}{L}\sum(x_i - \hat{\mu}_{\mathbf{x}})^2$, maximizing this expression with respect to the model parameters $(\mu_k, \sigma_k)$ becomes equivalent to minimizing a function of the differences $(\hat{\mu}_{\mathbf{x}} - \mu_k)$ and $(\hat{\sigma}_{\mathbf{x}}^2 - \sigma_k^2)$. To relate this expression to the sample statistics of the window $\mathbf{x}$, we first decompose the summation term by adding and subtracting the sample mean $\hat{\mu}_{\mathbf{x}}$:

$$\sum_{i=1}^{L}(x_i - \mu_k)^2 = \sum_{i=1}^{L}((x_i - \hat{\mu}_{\mathbf{x}}) + (\hat{\mu}_{\mathbf{x}} - \mu_k))^2$$

$$= \sum_{i=1}^{L}(x_i - \hat{\mu}_{\mathbf{x}})^2 + \sum_{i=1}^{L}(\hat{\mu}_{\mathbf{x}} - \mu_k)^2 + 2(\hat{\mu}_{\mathbf{x}} - \mu_k)\sum_{i=1}^{L}(x_i - \hat{\mu}_{\mathbf{x}}) \quad (18)$$

By the definition of the sample mean $\hat{\mu}_{\mathbf{x}}$, the cross-term $\sum_{i=1}^{L}(x_i - \hat{\mu}_{\mathbf{x}})$ is equal to zero. Using the definition of the sample variance $\hat{\sigma}_{\mathbf{x}}^2 = \frac{1}{L}\sum(x_i - \hat{\mu}_{\mathbf{x}})^2$, the summation simplifies to:

$$\sum_{i=1}^{L}(x_i - \mu_k)^2 = L \cdot \hat{\sigma}_{\mathbf{x}}^2 + L \cdot (\hat{\mu}_{\mathbf{x}} - \mu_k)^2 \quad (19)$$

Substituting this back into the log-likelihood, maximizing $\log p(\mathbf{x}|C_k)$ is equivalent to minimizing its negative $-\log p(\mathbf{x}|C_k)$. We can write the cost function $l(k)$ to minimize, by dropping constant terms such as $L/2$ and $\log(2\pi)$:

$$l(k) \propto \log(\sigma_k^2) + \frac{1}{\sigma_k^2}\left(L \cdot \hat{\sigma}_{\mathbf{x}}^2 + L \cdot (\hat{\mu}_{\mathbf{x}} - \mu_k)^2\right)$$

$$\propto \log(\sigma_k^2) + \frac{\hat{\sigma}_{\mathbf{x}}^2}{\sigma_k^2} + \frac{(\hat{\mu}_{\mathbf{x}} - \mu_k)^2}{\sigma_k^2} \quad (20)$$

$l(k)$ is the precise function to be minimized. This derivation confirms that maximizing the likelihood is equivalent to minimizing a function based on the differences between the sample statistics ($\hat{\mu}_{\mathbf{x}}$, $\hat{\sigma}_{\mathbf{x}}^2$) and the concept parameters ($\mu_k$, $\sigma_k^2$). While not identical, the Euclidean distance $d(\mathbf{g}_{\mathbf{x}}, \mathbf{g}_k) = \sqrt{(\hat{\mu}_{\mathbf{x}} - \mu_k)^2 + (\hat{\sigma}_{\mathbf{x}} - \sigma_k)^2}$ serves as a computationally efficient and effective proxy for this likelihood-based classification. It directly measures the dissimilarity between the observed sample's statistics and the learned concept's statistics. Minimizing this distance is analogous to finding the concept that provides the closest fit to the data, allowing for a rapid and decisive assignment. This approach ensures **conceptual purity**, each forecaster is trained only on data identified as belonging to its specific concept, which is critical for accurate specialization and effective knowledge retention in the face of recurring concept drifts.

## B.4   DERIVATION OF ONLINE GENE UPDATE

Equation 6 provides an efficient online method to update the global gene $\tilde{\mathbf{g}} = (\tilde{\mathbf{g}}_\mu, \tilde{\mathbf{g}}_\sigma)$, which represents the running mean and standard deviation of all instance genes encountered by a forecaster so far. This approach avoids storing all past data, making it highly suitable for online scenarios. Here, we provide a step-by-step derivation of these standard online update formulas.

**Mean Update Derivation.**   Let $\tilde{\mathbf{g}}_{\mu,n}$ be the mean of the first $n$ instance gene means, $\{\mathbf{g}_{x,\mu}^{(1)}, \mathbf{g}_{x,\mu}^{(2)}, \ldots, \mathbf{g}_{x,\mu}^{(n)}\}$. By definition:

$$\tilde{\mathbf{g}}_{\mu,n} = \frac{1}{n}\sum_{i=1}^{n}\mathbf{g}_{x,\mu}^{(i)} \quad (21)$$

When a new instance gene with mean $\mathbf{g}_{x,\mu}^{(n+1)}$ arrives, the new global mean $\tilde{\mathbf{g}}_{\mu,n+1}$ is calculated as:

$$\tilde{\mathbf{g}}_{\mu,n+1} = \frac{1}{n+1} \sum_{i=1}^{n+1} \mathbf{g}_{x,\mu}^{(i)} \tag{22}$$

$$= \frac{1}{n+1} \left( \left( \sum_{i=1}^{n} \mathbf{g}_{x,\mu}^{(i)} \right) + \mathbf{g}_{x,\mu}^{(n+1)} \right) \tag{23}$$

$$= \frac{1}{n+1} \left( n \cdot \left( \frac{1}{n} \sum_{i=1}^{n} \mathbf{g}_{x,\mu}^{(i)} \right) + \mathbf{g}_{x,\mu}^{(n+1)} \right) \tag{24}$$

$$= \frac{n \cdot \tilde{\mathbf{g}}_{\mu,n} + \mathbf{g}_{x,\mu}^{(n+1)}}{n+1} \tag{25}$$

This recursive formula allows us to compute the new mean using only the previous mean, the new value, and the sample count $n$. This corresponds to the first component of Equation 6.

**Variance Update Derivation.** The variance update is derived using a similar recursive approach, often known as a form of Welford's algorithm. Let $\tilde{\mathbf{g}}_{\sigma,n}^2$ be the variance of the first $n$ instance gene means. Let $M_{1,n} = \tilde{\mathbf{g}}_{\mu,n}$ be the first raw moment (the mean), and $M_{2,n} = \frac{1}{n} \sum_{i=1}^{n} (\mathbf{g}_{x,\mu}^{(i)})^2$ be the second raw moment. The variance is given by $\tilde{\mathbf{g}}_{\sigma,n}^2 = M_{2,n} - M_{1,n}^2$.

The moments can be updated online:

$$M_{1,n+1} = \frac{n \cdot M_{1,n} + \mathbf{g}_{x,\mu}^{(n+1)}}{n+1} \tag{26}$$

$$M_{2,n+1} = \frac{n \cdot M_{2,n} + (\mathbf{g}_{x,\mu}^{(n+1)})^2}{n+1} \tag{27}$$

Now, we derive the recursive formula for the variance $\tilde{\mathbf{g}}_{\sigma,n+1}^2$:

$$\tilde{\mathbf{g}}_{\sigma,n+1}^2 = M_{2,n+1} - M_{1,n+1}^2 \tag{28}$$

$$= \frac{nM_{2,n} + (\mathbf{g}_{x,\mu}^{(n+1)})^2}{n+1} - \left( \frac{nM_{1,n} + \mathbf{g}_{x,\mu}^{(n+1)}}{n+1} \right)^2 \tag{29}$$

$$= \frac{(n+1)(nM_{2,n} + (\mathbf{g}_{x,\mu}^{(n+1)})^2) - (n^2 M_{1,n}^2 + 2nM_{1,n}\mathbf{g}_{x,\mu}^{(n+1)} + (\mathbf{g}_{x,\mu}^{(n+1)})^2)}{(n+1)^2} \tag{30}$$

Substituting $M_{2,n} = \tilde{\mathbf{g}}_{\sigma,n}^2 + M_{1,n}^2$ into the expression:

$$= \frac{n(n+1)(\tilde{\mathbf{g}}_{\sigma,n}^2 + M_{1,n}^2) + n(\mathbf{g}_{x,\mu}^{(n+1)})^2 - n^2 M_{1,n}^2 - 2nM_{1,n}\mathbf{g}_{x,\mu}^{(n+1)}}{(n+1)^2} \tag{31}$$

$$= \frac{n(n+1)\tilde{\mathbf{g}}_{\sigma,n}^2 + (n^2+n)M_{1,n}^2 - n^2 M_{1,n}^2 + n(\mathbf{g}_{x,\mu}^{(n+1)})^2 - 2nM_{1,n}\mathbf{g}_{x,\mu}^{(n+1)}}{(n+1)^2} \tag{32}$$

$$= \frac{n(n+1)\tilde{\mathbf{g}}_{\sigma,n}^2 + nM_{1,n}^2 - 2nM_{1,n}\mathbf{g}_{x,\mu}^{(n+1)} + n(\mathbf{g}_{x,\mu}^{(n+1)})^2}{(n+1)^2} \tag{33}$$

$$= \frac{n(n+1)\tilde{\mathbf{g}}_{\sigma,n}^2 + n(M_{1,n} - \mathbf{g}_{x,\mu}^{(n+1)})^2}{(n+1)^2} \tag{34}$$

$$= \frac{n}{n+1}\tilde{\mathbf{g}}_{\sigma,n}^2 + \frac{n}{(n+1)^2}(\tilde{\mathbf{g}}_{\mu,n} - \mathbf{g}_{x,\mu}^{(n+1)})^2 \tag{35}$$

This expression is a numerically stable and efficient way to update the variance. This ensures that the global gene accurately reflects the statistical properties of all data seen by its corresponding forecaster.

### B.5 FORMAL REGRET ANALYSIS

To formally ground the effectiveness of CEP, we provide a high-level regret analysis. Traditional regret compares an online algorithm's cumulative loss to that of the best single fixed model in hindsight. However, this benchmark is ill-suited for concept drift scenarios, as no single model can perform optimally across all concepts. A more meaningful benchmark is an *oracle* that knows the true underlying concept at each time step and uses the optimal specialized forecaster for that concept.

**Problem Formulation.** Let the set of distinct, underlying concepts be $\mathcal{C} = \{C_1, \ldots, C_K\}$, where each concept $C_k$ is associated with a stationary data distribution $p_k(\mathbf{x}, \mathbf{y})$. For each concept $C_k$, there exists an optimal forecaster, $f_k^* = \arg\min_f \mathbb{E}_{(\mathbf{x}, \mathbf{y}) \sim p_k}[\mathcal{L}(f(\mathbf{x}), \mathbf{y})]$. At each time step $t$, the incoming data $(\mathbf{x}_t, \mathbf{y}_t)$ is generated from a distribution corresponding to the active concept $C(t) \in \mathcal{C}$. The oracle's cumulative loss over a horizon $T$ is $\sum_{t=1}^{T} \mathcal{L}_t(f_{C(t)}^*)$.

The regret of CEP, $R_T^{\text{CEP}}$, is the difference between its cumulative loss and the oracle's:

$$R_T^{\text{CEP}} = \sum_{t=1}^{T} \mathcal{L}_t(f_{\text{sel}}(t)) - \sum_{t=1}^{T} \mathcal{L}_t(f_{C(t)}^*) \tag{36}$$

where $f_{\text{sel}}(t)$ is the forecaster selected by CEP at time $t$.

**Decomposition of Regret.** The total regret can be decomposed into two primary components. Let $f_{\text{CEP}}(t)$ be the forecaster within CEP's pool that is designated for the true concept $C(t)$.

$$R_T^{\text{CEP}} = \underbrace{\sum_{t=1}^{T} \left(\mathcal{L}_t(f_{\text{sel}}(t)) - \mathcal{L}_t(f_{\text{CEP}}(t))\right)}_{\text{Identification Regret}} + \underbrace{\sum_{t=1}^{T} \left(\mathcal{L}_t(f_{\text{CEP}}(t)) - \mathcal{L}_t(f_{C(t)}^*)\right)}_{\text{Estimation Regret}} \tag{37}$$

- **Identification Regret**: This term captures the loss incurred from selecting the wrong forecaster (i.e., $f_{\text{sel}}(t) \neq f_{\text{CEP}}(t)$). This occurs if the gene of an instance from concept $C_k$ is closer to the gene of another forecaster $f_j$. The core strength of CEP lies in minimizing this regret. As long as the statistical genes $\mathbf{g}_k$ of different concepts are well-separated in the feature space, the probability of misidentification is low. This regret is bounded by the number of misidentification events, which is minimal in scenarios with distinct concept shifts.
- **Estimation Regret**: This term reflects the cumulative loss because our online-trained forecasters are not the true optimal ones. CEP's key innovation is converting a single non-stationary learning problem into a set of parallel, stationary subproblems. Each forecaster $f_k$ in the pool is updated only with data from its corresponding concept $C_k$. For standard online learning algorithms (like online gradient descent) on a stationary distribution, the regret is known to be sublinear in the number of samples received, e.g., $O(\sqrt{T_k})$ (Zinkevich, 2003; Hazan et al., 2016) where $T_k$ is the total number of instances seen for concept $C_k$. The total estimation regret is the sum of these sublinear regrets across all active forecasters.

By design, CEP excels at minimizing the Identification Regret through its statistically-grounded hard assignment mechanism. It effectively bounds the Estimation Regret by ensuring each specialized forecaster learns on a stable data distribution. Consequently, the total regret of CEP is expected to grow sublinearly with time, demonstrating its convergence towards the optimal dynamic strategy. This provides a strong theoretical justification for its superior performance over single-model approaches that suffer from catastrophic forgetting, which often leads to linear regret in concept drift settings.

## C EXPERIMENT DETAILS

### C.1 DATASET

During the experiments, we used a diverse range of datasets to comprehensively assess the time series forecasting model. **ETT**, which covers two years, monitors the oil temperature and six

power load characteristics. ETTh2 logs data hourly, while ETTm1 records at 15-minute intervals, presenting multiple granularities for the analysis of electricity-related time series. **Electricity (ECL)** dataset, which monitors the electricity consumption of 321 clients from 2012 to 2014, facilitates the exploration of intricate consumption patterns. **Weather (WTH)** monitored 11 climatic aspects at around 1,600 spots throughout the United States every hour from 2010 to 2013. **Traffic**, sourced from the California Department of Transportation, supplies hourly freeway occupancy rates in the San Francisco Bay Area, which is beneficial for traffic flow forecasting. **Exchange**, spanning from 1990 to 2016 and containing the daily exchange rates of eight countries, is instrumental in predicting the fluctuations in the foreign exchange market (Wu et al., 2021). All the datasets were used with a single channel with the recurring concept shift in Table 5. In the experiment, the ratio of the data set in the warm-up stage and online stage is 25 : 75.

Table 5: Dataset statistics.

| Data | ECL | ETTh1 | ETTh2 | ETTm1 | ETTm2 | Exchange | Traffic | WTH |
|------|-----|-------|-------|-------|-------|----------|---------|-----|
| Feature | MT_104 | LULL | LUFL | LULL | MUFL | OT | 16 | DewPointFarenheit |
| Timestep | 26,304 | 14,400 | 14,400 | 14,400 | 14,400 | 7,588 | 17,544 | 35,064 |

## C.2 EXPERIMENTAL SETTING

The experiments were conducted on a computer equipped with an AMD Ryzen 9 7950X CPU, which boasts 16 cores and a clock speed of 5.1 GHz. This computer is furnished with 64 GB of memory, and all experiments were executed using PyTorch. The default hyperparameters of `CEP` are presented in Table 6. The details regarding the hyperparameter sensitivity are provided in Section C.6.

Table 6: Hyparameters of `CEP` in the experiments.

| Hyperparameter | $\tau_\mu$ | $\tau_\sigma$ | $\tau_g$ | $\tau_l$ | $\tau_{safe}$ | $\tau_e$ |
|----------------|-----------|---------------|----------|----------|---------------|----------|
| Value | 3.0 | / | 0.8 | 0.2 | 15 | 1.5 |

### C.2.1 GRADUAL CONCEPT DRIFT

To assess model adaptability to gradual distribution shifts, we set the forecast horizon to $\mathbf{H} = 1$. Our results show that no single online method consistently excels. A naive model achieved state-of-the-art (SOTA) results on four datasets, matching the performance of an experience replay-based method (Buzzega et al., 2020). Other methods like FSNet (Pham et al., 2023) and OneNet (Zhang et al., 2023) proved less effective in this gradual shift scenario. Given that all models produced low and acceptable errors, we conclude that sophisticated techniques for handling gradual concept drift are unnecessary for short-term forecasting.

When the forecasting horizon $\mathbf{H} = 1$, each input instance updates only one time step. This leads to only minor alterations in the overall input data, resulting in a gradual concept shift. We conducted experiments with $\mathbf{H} = 1$ using previous online methods and naive forecasters with different common backbones. Additionally, we applied the `CEP` to the naive forecasters. The specific experimental results are presented in Table 7. From the experimental results, it is evident that the online time series forecasting methods did not demonstrate an absolute advantage over the naive forecasters. Both achieved state-of-the-art (SOTA) performance in four datasets.

## C.3 FULL EXPERIMENT RESULT OF NAIVE FORECASTERS

We conducted experiments with forecasting horizons set to $\mathbf{H} \in \{30, 60\}$. In each instance, at least half of the total input length is updated, preventing a gradual shift in the distribution. Consequently, the phenomenon of concept recurrence becomes more pronounced. The detailed experimental results are presented in Table 8. These results demonstrate that `CEP` substantially enhances the prediction performance of the naive forecaster. To more intuitively illustrate the forecasting effect of `CEP`, the forecasting results of `CEP` and the naive forecasters are depicted in Figure 8.

Table 7: Full MSE errors of gradual concept shift experiments. Enhanced and reduced outcomes are marked with █ and █ respectively. The **best** and second-best performances are highlighted.

| Data | ECL | | ETTh1 | | ETTh2 | | ETTm1 | | ETTm2 | | Exchange | | Traffic | | WTH | |
|---|---|---|---|---|---|---|---|---|---|---|---|---|---|---|---|---|
| | MSE | Std | MSE | Std | MSE | Std | MSE | Std | MSE | Std | MSE | Std | MSE | Std | MSE | Std |
| ER | 0.124 | 0.004 | 0.084 | 0.001 | 0.693 | 0.006 | 0.089 | 0.000 | 0.050 | 0.000 | 0.189 | 0.025 | 0.606 | 0.361 | 0.091 | 0.015 |
| DER++ | 0.107 | 0.004 | 0.081 | 0.001 | **0.663** | 0.006 | **0.086** | 0.000 | **0.048** | 0.000 | 0.168 | 0.020 | 0.507 | 0.321 | 0.072 | 0.012 |
| FSNet | 0.208 | 0.026 | 0.091 | 0.001 | 0.767 | 0.010 | 0.123 | 0.009 | 0.062 | 0.001 | 1.175 | 0.425 | 0.618 | 0.065 | 0.096 | 0.004 |
| OneNet | 0.109 | 0.004 | 0.084 | 0.001 | 0.737 | 0.021 | 0.115 | 0.005 | 0.058 | 0.002 | 0.248 | 0.321 | 0.303 | 0.015 | 0.065 | 0.002 |
| TCN | 0.070 | 0.003 | **0.079** | 0.001 | 0.701 | 0.006 | 0.094 | 0.001 | 0.053 | 0.001 | **0.033** | 0.001 | **0.190** | 0.011 | **0.045** | 0.000 |
| +CEP ⧖ | 0.070 | 0.003 | **0.079** | 0.001 | 0.701 | 0.006 | 0.094 | 0.001 | 0.053 | 0.001 | **0.033** | 0.001 | **0.190** | 0.011 | **0.045** | 0.000 |
| TimesNet | 0.096 | 0.009 | 0.110 | 0.010 | 0.888 | 0.045 | 0.185 | 0.012 | 0.093 | 0.010 | 0.077 | 0.009 | 0.329 | 0.028 | 0.077 | 0.005 |
| +CEP ⧖ | 0.096 | 0.009 | 0.110 | 0.010 | 0.901 | 0.071 | 0.192 | 0.014 | 0.093 | 0.010 | 0.077 | 0.008 | 0.329 | 0.028 | 0.077 | 0.005 |
| DLinear | 0.160 | 0.000 | 0.109 | 0.000 | 1.026 | 0.001 | 0.218 | 0.000 | 0.069 | 0.000 | 0.077 | 0.000 | 0.251 | 0.000 | 0.061 | 0.000 |
| +CEP ⧖ | 0.160 | 0.000 | 0.109 | 0.000 | 1.026 | 0.001 | 0.223 | 0.001 | 0.069 | 0.000 | 0.077 | 0.000 | 0.251 | 0.000 | 0.061 | 0.000 |
| PatchTST | 0.105 | 0.013 | 0.096 | 0.007 | 0.966 | 0.108 | 0.142 | 0.019 | 0.068 | 0.004 | 0.066 | 0.009 | 0.284 | 0.039 | 0.072 | 0.017 |
| +CEP ⧖ | 0.105 | 0.013 | 0.095 | 0.006 | 0.993 | 0.102 | 0.151 | 0.026 | 0.071 | 0.008 | 0.067 | 0.007 | 0.284 | 0.039 | 0.067 | 0.015 |
| iTransformer | 0.126 | 0.033 | 0.117 | 0.009 | 0.956 | 0.042 | 0.239 | 0.047 | 0.089 | 0.007 | 0.083 | 0.010 | 0.308 | 0.015 | 0.108 | 0.015 |
| +CEP ⧖ | 0.123 | 0.023 | 0.116 | 0.009 | 0.950 | 0.050 | 0.221 | 0.052 | 0.089 | 0.007 | 0.095 | 0.029 | 0.305 | 0.012 | 0.108 | 0.015 |
| TimeMixer | **0.066** | 0.005 | 0.091 | 0.005 | 0.701 | 0.018 | 0.113 | 0.007 | 0.060 | 0.002 | 0.040 | 0.004 | 0.263 | 0.016 | 0.050 | 0.003 |
| +CEP ⧖ | **0.066** | 0.005 | 0.091 | 0.005 | 0.702 | 0.016 | 0.114 | 0.007 | 0.060 | 0.002 | 0.039 | 0.003 | 0.263 | 0.016 | 0.051 | 0.002 |

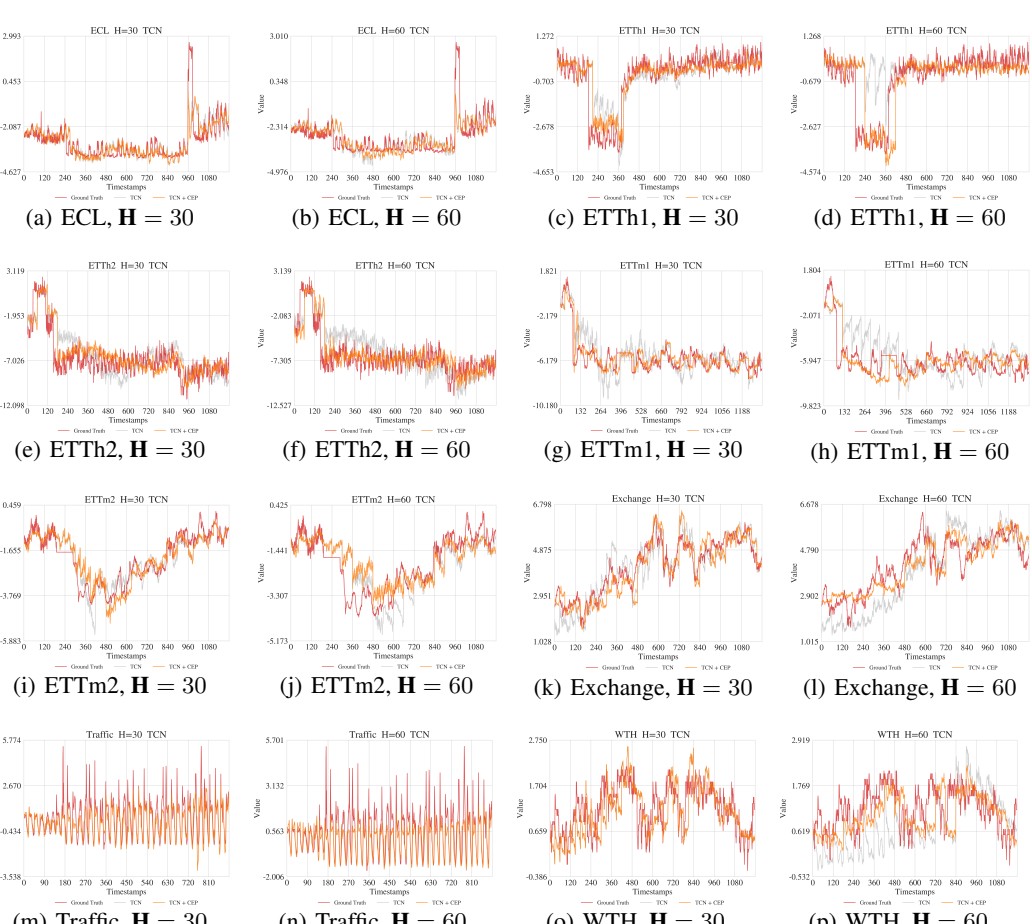

(a) ECL, $\mathbf{H} = 30$  (b) ECL, $\mathbf{H} = 60$  (c) ETTh1, $\mathbf{H} = 30$  (d) ETTh1, $\mathbf{H} = 60$

(e) ETTh2, $\mathbf{H} = 30$  (f) ETTh2, $\mathbf{H} = 60$  (g) ETTm1, $\mathbf{H} = 30$  (h) ETTm1, $\mathbf{H} = 60$

(i) ETTm2, $\mathbf{H} = 30$  (j) ETTm2, $\mathbf{H} = 60$  (k) Exchange, $\mathbf{H} = 30$  (l) Exchange, $\mathbf{H} = 60$

(m) Traffic, $\mathbf{H} = 30$  (n) Traffic, $\mathbf{H} = 60$  (o) WTH, $\mathbf{H} = 30$  (p) WTH, $\mathbf{H} = 60$

Figure 8: Visualization of results by naive forecasters *vs.* forecasters with `CEP` across all datasets

## C.4 FULL EXPERIMENT RESULT OF ONLINE METHODS

All online methods incorporate the same TCN component. Therefore, the performance of online methods, the naive TCN, and the TCN utilizing the `CEP` is compared that the full experimental results are presented in Table 9. An intriguing phenomenon is that, in comparison to the naive TCN, online methods nearly fail in most scenarios. A plausible explanation is that, owing to delayed feedback, the information acquired by online methods is delayed and fails to adapt promptly to concept changes.

Table 8: Full MSE errors of different forecasters when using CEP are presented. Enhanced and reduced outcomes are marked with ▢ and ▢ respectively.

**MSE**

| Data | ECL 30 | ECL 60 | ETTh1 30 | ETTh1 60 | ETTh2 30 | ETTh2 60 | ETTm1 30 | ETTm1 60 | ETTm2 30 | ETTm2 60 | Exchange 30 | Exchange 60 | Traffic 30 | Traffic 60 | WTH 30 | WTH 60 |
|---|---|---|---|---|---|---|---|---|---|---|---|---|---|---|---|---|
| TimeMixer | 0.266 | 0.314 | 0.295 | 0.376 | 2.112 | 3.149 | 0.577 | 0.933 | 0.219 | 0.255 | 0.245 | 0.466 | 0.433 | 0.524 | 0.313 | 0.399 |
| +CEP | 0.244 | 0.297 | 0.274 | 0.371 | 2.000 | 3.016 | 0.579 | 0.895 | 0.216 | 0.254 | 0.233 | 0.450 | 0.433 | 0.524 | 0.313 | 0.396 |
|  | -8.34% | -5.17% | -7.13% | -1.38% | -5.29% | -4.21% | 0.38% | -4.10% | -1.19% | -0.39% | -5.13% | -3.43% | 0.00% | 0.00% | 0.00% | -0.55% |
| iTransformer | 0.254 | 0.296 | 0.280 | 0.371 | 1.974 | 3.159 | 0.606 | 0.915 | 0.212 | 0.267 | 0.274 | 0.495 | 0.437 | 0.534 | 0.311 | 0.395 |
| +CEP | 0.229 | 0.282 | 0.275 | 0.372 | 1.968 | 3.116 | 0.594 | 0.886 | 0.210 | 0.265 | 0.249 | 0.484 | 0.437 | 0.534 | 0.311 | 0.394 |
|  | -9.62% | -4.72% | -2.00% | 0.11% | -0.27% | -1.34% | -1.98% | -3.08% | -1.22% | -0.45% | -9.33% | -2.18% | 0.00% | 0.00% | 0.00% | -0.40% |
| PatchTST | 0.279 | 0.340 | 0.284 | 0.380 | 2.028 | 3.160 | 0.607 | 0.911 | 0.209 | 0.265 | 0.263 | 0.479 | 0.582 | 0.589 | 0.310 | 0.398 |
| +CEP | 0.265 | 0.328 | 0.276 | 0.377 | 1.922 | 2.995 | 0.567 | 0.875 | 0.208 | 0.264 | 0.234 | 0.467 | 0.582 | 0.589 | 0.310 | 0.397 |
|  | -5.01% | -3.70% | -2.81% | -0.95% | -5.26% | -5.22% | -6.55% | -3.97% | -0.67% | -0.45% | -11.25% | -2.51% | 0.00% | 0.00% | 0.00% | -0.40% |
| DLinear | 0.275 | 0.340 | 0.289 | 0.385 | 2.070 | 3.366 | 0.656 | 1.042 | 0.200 | 0.271 | 0.292 | 0.523 | 0.614 | 0.641 | 0.295 | 0.388 |
| +CEP | 0.264 | 0.335 | 0.259 | 0.356 | 2.052 | 3.209 | 0.586 | 0.917 | 0.196 | 0.259 | 0.261 | 0.494 | 0.614 | 0.641 | 0.295 | 0.386 |
|  | -4.07% | -1.41% | -10.38% | -7.43% | -0.88% | -4.65% | -10.69% | -12.05% | -2.00% | -4.36% | -10.69% | -5.62% | 0.00% | 0.00% | 0.00% | -0.62% |
| SegRNN | 0.242 | 0.314 | 0.250 | 0.356 | 1.864 | 3.391 | 0.554 | 0.883 | 0.201 | 0.263 | 0.193 | 0.397 | 0.693 | 0.930 | 0.298 | 0.384 |
| +CEP | 0.239 | 0.312 | 0.248 | 0.353 | 1.840 | 3.328 | 0.551 | 0.881 | 0.197 | 0.259 | 0.192 | 0.395 | 0.693 | 0.930 | 0.298 | 0.385 |
|  | -1.08% | -0.57% | -0.80% | -0.73% | -1.26% | -1.88% | -0.58% | -0.16% | -1.89% | -1.82% | -0.52% | -0.50% | 0.00% | 0.00% | 0.00% | 0.36% |
| TimesNet | 0.281 | 0.330 | 0.304 | 0.398 | 2.141 | 3.212 | 0.692 | 0.981 | 0.237 | 0.279 | 0.297 | 0.546 | 0.492 | 0.597 | 0.323 | 0.406 |
| +CEP | 0.259 | 0.319 | 0.291 | 0.390 | 2.054 | 3.065 | 0.639 | 0.947 | 0.234 | 0.277 | 0.259 | 0.530 | 0.492 | 0.597 | 0.323 | 0.404 |
|  | -7.77% | -3.27% | -4.08% | -2.11% | -4.04% | -4.59% | -7.60% | -3.53% | -1.35% | -0.79% | -13.05% | -3.07% | 0.00% | 0.00% | 0.00% | -0.49% |
| TCN | 0.295 | 0.393 | 0.371 | 0.555 | 2.493 | 3.838 | 0.761 | 1.201 | 0.221 | 0.320 | 0.389 | 0.694 | 0.668 | 0.714 | 0.314 | 0.439 |
| +CEP | 0.261 | 0.358 | 0.392 | 0.526 | 2.164 | 3.443 | 0.638 | 0.964 | 0.219 | 0.309 | 0.272 | 0.553 | 0.668 | 0.714 | 0.314 | 0.450 |
|  | -11.71% | -9.00% | 5.88% | -5.19% | -13.20% | -10.29% | -16.21% | -19.79% | -0.81% | -3.62% | -30.25% | -20.33% | 0.00% | 0.00% | 0.00% | 2.46% |

**Std**

| Data | ECL 30 | ECL 60 | ETTh1 30 | ETTh1 60 | ETTh2 30 | ETTh2 60 | ETTm1 30 | ETTm1 60 | ETTm2 30 | ETTm2 60 | Exchange 30 | Exchange 60 | Traffic 30 | Traffic 60 | WTH 30 | WTH 60 |
|---|---|---|---|---|---|---|---|---|---|---|---|---|---|---|---|---|
| TimeMixer | 0.006 | 0.003 | 0.016 | 0.006 | 0.060 | 0.032 | 0.027 | 0.019 | 0.020 | 0.018 | 0.010 | 0.013 | 0.018 | 0.033 | 0.002 | 0.003 |
| +CEP | 0.007 | 0.003 | 0.005 | 0.005 | 0.013 | 0.006 | 0.025 | 0.010 | 0.022 | 0.018 | 0.011 | 0.014 | 0.018 | 0.033 | 0.002 | 0.003 |
| iTransformer | 0.004 | 0.003 | 0.002 | 0.004 | 0.014 | 0.027 | 0.010 | 0.012 | 0.002 | 0.003 | 0.010 | 0.008 | 0.012 | 0.009 | 0.002 | 0.001 |
| +CEP | 0.004 | 0.001 | 0.002 | 0.004 | 0.009 | 0.036 | 0.012 | 0.020 | 0.002 | 0.003 | 0.003 | 0.009 | 0.012 | 0.009 | 0.002 | 0.002 |
| PatchTST | 0.004 | 0.002 | 0.001 | 0.005 | 0.018 | 0.035 | 0.014 | 0.012 | 0.005 | 0.007 | 0.002 | 0.011 | 0.006 | 0.004 | 0.001 | 0.001 |
| +CEP | 0.002 | 0.004 | 0.004 | 0.004 | 0.015 | 0.019 | 0.007 | 0.008 | 0.005 | 0.007 | 0.006 | 0.012 | 0.006 | 0.004 | 0.001 | 0.001 |
| DLinear | 0.000 | 0.001 | 0.001 | 0.002 | 0.002 | 0.002 | 0.002 | 0.001 | 0.001 | 0.006 | 0.012 | 0.013 | 0.001 | 0.000 | 0.000 | 0.000 |
| +CEP | 0.001 | 0.000 | 0.000 | 0.001 | 0.003 | 0.003 | 0.002 | 0.001 | 0.001 | 0.006 | 0.010 | 0.013 | 0.001 | 0.000 | 0.000 | 0.000 |
| SegRNN | 0.003 | 0.003 | 0.001 | 0.002 | 0.006 | 0.011 | 0.001 | 0.004 | 0.003 | 0.002 | 0.001 | 0.000 | 0.013 | 0.026 | 0.002 | 0.001 |
| +CEP | 0.003 | 0.003 | 0.001 | 0.002 | 0.004 | 0.006 | 0.002 | 0.003 | 0.003 | 0.002 | 0.001 | 0.000 | 0.013 | 0.026 | 0.002 | 0.002 |
| TimesNet | 0.012 | 0.008 | 0.010 | 0.020 | 0.020 | 0.042 | 0.062 | 0.045 | 0.004 | 0.009 | 0.023 | 0.039 | 0.028 | 0.036 | 0.005 | 0.003 |
| +CEP | 0.014 | 0.011 | 0.009 | 0.014 | 0.059 | 0.066 | 0.022 | 0.048 | 0.003 | 0.008 | 0.019 | 0.052 | 0.028 | 0.036 | 0.006 | 0.003 |
| TCN | 0.009 | 0.007 | 0.013 | 0.058 | 0.053 | 0.107 | 0.006 | 0.052 | 0.021 | 0.012 | 0.019 | 0.016 | 0.070 | 0.006 | 0.002 | 0.003 |
| +CEP | 0.010 | 0.005 | 0.015 | 0.044 | 0.021 | 0.083 | 0.006 | 0.031 | 0.019 | 0.011 | 0.009 | 0.018 | 0.070 | 0.006 | 0.002 | 0.004 |

The CEP significantly enhances the performance of TCN, effectively reducing prediction errors across a wide range of datasets.

Table 9: Full MSE error of previous online forecasting methods. The **best** and second-best performances are highlighted.

**MSE**

| Data | ECL 30 | ECL 60 | ETTh1 30 | ETTh1 60 | ETTh2 30 | ETTh2 60 | ETTm1 30 | ETTm1 60 | ETTm2 30 | ETTm2 60 | Exchange 30 | Exchange 60 | Traffic 30 | Traffic 60 | WTH 30 | WTH 60 |
|---|---|---|---|---|---|---|---|---|---|---|---|---|---|---|---|---|
| TCN | 0.295 | 0.393 | 0.371 | 0.555 | 2.493 | 3.838 | 0.761 | 1.201 | 0.221 | 0.320 | 0.389 | 0.694 | 0.668 | 0.714 | 0.314 | 0.439 |
| ER | 0.563 | 0.685 | 0.358 | 0.551 | 2.624 | 4.035 | 0.901 | 1.862 | 0.246 | 0.399 | 2.268 | 3.550 | 1.359 | 0.862 | 0.524 | 0.592 |
| DER++ | 0.538 | 0.621 | 0.332 | 0.481 | 2.339 | 3.806 | 0.932 | 1.622 | 0.235 | 0.373 | 1.568 | 3.462 | 1.325 | 0.827 | 0.486 | 0.573 |
| FSNet | 0.863 | 0.822 | 0.392 | 0.547 | 3.118 | 5.025 | 1.111 | 2.013 | 0.275 | 0.419 | 3.093 | 3.879 | 1.618 | 1.314 | 0.641 | 0.846 |
| OneNet | 0.385 | 0.466 | 0.344 | 0.475 | 2.540 | 4.265 | 0.949 | 1.237 | 0.237 | 0.332 | 0.899 | 1.361 | 0.705 | 0.640 | 0.360 | 0.469 |
| CEP | 0.261 | 0.358 | 0.392 | 0.526 | 2.164 | 3.443 | 0.638 | 0.964 | 0.219 | 0.309 | 0.272 | 0.553 | 0.668 | 0.714 | 0.314 | 0.450 |

**Std**

| Data | ECL 30 | ECL 60 | ETTh1 30 | ETTh1 60 | ETTh2 30 | ETTh2 60 | ETTm1 30 | ETTm1 60 | ETTm2 30 | ETTm2 60 | Exchange 30 | Exchange 60 | Traffic 30 | Traffic 60 | WTH 30 | WTH 60 |
|---|---|---|---|---|---|---|---|---|---|---|---|---|---|---|---|---|
| TCN | 0.010 | 0.005 | 0.015 | 0.044 | 0.021 | 0.083 | 0.006 | 0.031 | 0.019 | 0.011 | 0.009 | 0.018 | 0.070 | 0.006 | 0.002 | 0.004 |
| ER | 0.006 | 0.050 | 0.002 | 0.026 | 0.116 | 0.040 | 0.016 | 0.058 | 0.006 | 0.005 | 0.243 | 0.684 | 0.283 | 0.048 | 0.014 | 0.012 |
| DER++ | 0.015 | 0.055 | 0.007 | 0.034 | 0.126 | 0.070 | 0.013 | 0.068 | 0.007 | 0.007 | 0.847 | 0.364 | 0.278 | 0.048 | 0.014 | 0.012 |
| FSNet | 0.108 | 0.044 | 0.014 | 0.027 | 0.042 | 0.088 | 0.008 | 0.060 | 0.019 | 0.012 | 0.335 | 0.508 | 0.093 | 0.147 | 0.055 | 0.042 |
| OneNet | 0.011 | 0.036 | 0.013 | 0.016 | 0.068 | 0.254 | 0.034 | 0.099 | 0.013 | 0.027 | 0.179 | 0.151 | 0.029 | 0.018 | 0.011 | 0.014 |
| CEP | 0.009 | 0.007 | 0.013 | 0.058 | 0.053 | 0.107 | 0.006 | 0.052 | 0.021 | 0.012 | 0.019 | 0.016 | 0.070 | 0.006 | 0.002 | 0.003 |

## C.5 Ablation Study

In the ablation experiment, we tested the mechanisms in the CEP. These mechanisms include evolution, using only fast genes, using only slow genes, elimination, inspiration, and gradient abandonment. We

tested all the forecasters and took the average of the results. The specific experimental results are in Table 10. The most basic evolution mechanism in the `CEP` played a major role when compared to the performance without `CEP`. Other mechanisms perform partially well for two reasons. First, the performance of different forecasters can vary under the influence of `CEP`, and second, the use of uniform default parameters limits the effectiveness of the datasets with different characteristics.

Table 10: Full ablation study results. All results are the averaged MSE of different backbone forecasters. The **best** and second-best performances are highlighted.

| Evolution | Inspiration | Gradient Abandonment | Elimination | Fast Gene | Slow Gene | ECL 30 | ECL 60 | ETTh1 30 | ETTh1 60 | ETTh2 30 | ETTh2 60 | ETTm1 30 | ETTm1 60 | ETTm2 30 | ETTm2 60 | Exchange 30 | Exchange 60 | Traffic 30 | Traffic 60 | WTH 30 | WTH 60 | AVG | Rank |
|---|---|---|---|---|---|---|---|---|---|---|---|---|---|---|---|---|---|---|---|---|---|---|---|
| | | | TCN | | | 0.295 | 0.398 | 0.359 | 0.512 | 2.480 | 3.732 | 0.753 | 1.228 | 0.208 | 0.313 | 0.361 | 0.687 | 0.696 | 0.724 | 0.313 | 0.437 | 0.844 | 25 |
| ✓ | ✗ | ✗ | ✓ | ✗ | ✓ | 0.275 | 0.377 | 0.332 | 0.477 | 2.339 | 3.845 | 0.627 | 0.929 | 0.219 | 0.309 | 0.361 | 0.687 | 0.696 | 0.724 | 0.313 | 0.437 | 0.809 | 24 |
| ✓ | ✗ | ✗ | ✓ | ✗ | ✓ | 0.294 | 0.392 | 0.332 | 0.446 | 2.339 | 3.814 | 0.627 | 0.929 | 0.219 | 0.309 | 0.361 | 0.687 | 0.696 | 0.724 | 0.313 | 0.437 | 0.807 | 23 |
| ✓ | ✗ | ✗ | ✓ | ✓ | ✗ | 0.271 | 0.373 | 0.330 | 0.443 | 2.368 | 3.958 | 0.627 | 0.988 | 0.207 | 0.309 | 0.288 | 0.579 | 0.696 | 0.724 | 0.312 | 0.415 | 0.806 | 22 |
| ✓ | ✗ | ✓ | ✓ | ✗ | ✓ | 0.275 | 0.379 | 0.348 | 0.480 | 2.353 | 3.688 | 0.650 | 0.935 | 0.210 | 0.305 | 0.361 | 0.687 | 0.696 | 0.724 | 0.313 | 0.437 | 0.803 | 21 |
| ✓ | ✗ | ✗ | ✗ | ✓ | ✗ | 0.281 | 0.383 | 0.330 | 0.443 | 2.323 | 3.886 | 0.627 | 0.988 | 0.204 | 0.309 | 0.291 | 0.567 | 0.696 | 0.724 | 0.307 | 0.410 | 0.798 | 20 |
| ✓ | ✗ | ✓ | ✗ | ✗ | ✗ | 0.296 | 0.408 | 0.348 | 0.462 | 2.353 | 3.565 | 0.650 | 0.935 | 0.217 | 0.305 | 0.361 | 0.687 | 0.696 | 0.724 | 0.313 | 0.437 | 0.797 | 19 |
| ✓ | ✗ | ✗ | ✗ | ✓ | ✓ | 0.289 | 0.378 | 0.330 | 0.443 | 2.393 | 3.815 | 0.627 | 0.929 | 0.213 | 0.309 | 0.298 | 0.525 | 0.696 | 0.724 | 0.313 | 0.434 | 0.795 | 18 |
| ✓ | ✗ | ✗ | ✓ | ✓ | ✓ | 0.274 | 0.377 | 0.330 | 0.443 | 2.366 | 3.832 | 0.627 | 0.929 | 0.207 | 0.309 | 0.292 | 0.525 | 0.696 | 0.724 | 0.313 | 0.434 | 0.792 | 17 |
| ✓ | ✓ | ✓ | ✓ | ✗ | ✓ | 0.270 | 0.363 | 0.397 | 0.505 | 2.204 | 3.604 | 0.628 | 0.973 | 0.212 | 0.302 | 0.361 | 0.687 | 0.696 | 0.724 | 0.313 | 0.437 | 0.792 | 16 |
| ✓ | ✓ | ✗ | ✓ | ✗ | ✗ | 0.268 | 0.370 | 0.379 | 0.490 | 2.314 | 3.641 | 0.596 | 1.059 | 0.208 | 0.307 | 0.267 | 0.564 | 0.696 | 0.724 | 0.310 | 0.422 | 0.788 | 15 |
| ✓ | ✗ | ✗ | ✓ | ✗ | ✓ | 0.271 | 0.358 | 0.373 | 0.493 | 2.160 | 3.651 | 0.596 | 0.936 | 0.215 | 0.307 | 0.361 | 0.687 | 0.696 | 0.724 | 0.313 | 0.437 | 0.786 | 14 |
| ✓ | ✓ | ✓ | ✓ | ✗ | ✗ | 0.281 | 0.392 | 0.331 | 0.447 | 2.311 | 3.665 | 0.650 | 1.026 | 0.205 | 0.305 | 0.291 | 0.534 | 0.696 | 0.724 | 0.307 | 0.411 | 0.786 | 13 |
| ✓ | ✗ | ✓ | ✓ | ✗ | ✗ | 0.297 | 0.394 | 0.373 | 0.489 | 2.160 | 3.550 | 0.596 | 0.936 | 0.208 | 0.307 | 0.361 | 0.687 | 0.696 | 0.724 | 0.313 | 0.437 | 0.783 | 12 |
| ✓ | ✗ | ✓ | ✓ | ✓ | ✗ | 0.269 | 0.378 | 0.331 | 0.447 | 2.353 | 3.577 | 0.650 | 1.026 | 0.207 | 0.305 | 0.287 | 0.545 | 0.696 | 0.724 | 0.313 | 0.416 | 0.783 | 11 |
| ✓ | ✓ | ✓ | ✗ | ✓ | ✓ | 0.298 | 0.412 | 0.397 | 0.506 | 2.204 | 3.368 | 0.628 | 0.973 | 0.205 | 0.305 | 0.361 | 0.687 | 0.696 | 0.724 | 0.313 | 0.437 | 0.782 | 10 |
| ✓ | ✗ | ✓ | ✗ | ✓ | ✗ | 0.286 | 0.387 | 0.331 | 0.447 | 2.334 | 3.651 | 0.650 | 0.935 | 0.207 | 0.305 | 0.288 | 0.518 | 0.696 | 0.724 | 0.313 | 0.438 | 0.782 | 9 |
| ✓ | ✓ | ✓ | ✓ | ✓ | ✗ | 0.264 | 0.378 | 0.380 | 0.495 | 2.313 | 3.431 | 0.628 | 1.102 | 0.209 | 0.302 | 0.269 | 0.533 | 0.696 | 0.724 | 0.310 | 0.424 | 0.779 | 8 |
| ✓ | ✓ | ✗ | ✗ | ✓ | ✗ | 0.278 | 0.371 | 0.379 | 0.490 | 2.255 | 3.526 | 0.596 | 1.059 | 0.205 | 0.307 | 0.272 | 0.561 | 0.696 | 0.724 | 0.302 | 0.414 | 0.777 | 7 |
| ✓ | ✗ | ✓ | ✗ | ✓ | ✗ | 0.277 | 0.383 | 0.380 | 0.495 | 2.255 | 3.425 | 0.628 | 1.102 | 0.206 | 0.302 | 0.274 | 0.526 | 0.696 | 0.724 | 0.303 | 0.414 | 0.774 | 6 |
| ✓ | ✗ | ✓ | ✓ | ✓ | ✓ | 0.271 | 0.380 | 0.331 | 0.447 | 2.318 | 3.570 | 0.650 | 0.935 | 0.208 | 0.307 | 0.283 | 0.518 | 0.696 | 0.724 | 0.313 | 0.438 | 0.774 | 5 |
| ✓ | ✓ | ✗ | ✓ | ✓ | ✓ | 0.273 | 0.359 | 0.379 | 0.490 | 2.165 | 3.592 | 0.596 | 0.936 | 0.208 | 0.307 | 0.266 | 0.542 | 0.696 | 0.724 | 0.313 | 0.440 | 0.768 | 4 |
| ✓ | ✓ | ✗ | ✓ | ✗ | ✓ | 0.286 | 0.377 | 0.379 | 0.490 | 2.191 | 3.531 | 0.596 | 0.936 | 0.208 | 0.307 | 0.269 | 0.542 | 0.696 | 0.724 | 0.313 | 0.440 | 0.768 | 3 |
| ✓ | ✓ | ✓ | ✗ | ✓ | ✓ | 0.281 | 0.389 | 0.380 | 0.495 | 2.187 | 3.406 | 0.628 | 0.973 | 0.206 | 0.302 | 0.266 | 0.546 | 0.696 | 0.724 | 0.313 | 0.445 | 0.765 | 2 |
| | | CEP ⚡ | | | | 0.265 | 0.363 | 0.380 | 0.495 | 2.174 | 3.424 | 0.628 | 0.973 | 0.209 | 0.302 | 0.262 | 0.546 | 0.696 | 0.724 | 0.313 | 0.445 | 0.762 | 1 |

## C.6 HYPERPARAMETER SENSITIVITY

We conducted a detailed hyperparameter sensitivity analysis to evaluate the impact of `CEP`'s core mechanisms. Experiments were performed on the ETTm1 dataset with a forecasting horizon of $H = 30$. The results, presented in Figure 9, provide insights into the system's behavior.

The analysis reveals that performance is most responsive to hyperparameters governing the lifecycle of forecasters. For the **Mean Threshold** ($\tau_\mu$), as shown in Figure 9(a), an optimal value exists around 2.0-3.0. A threshold that is too low causes excessive and unstable evolution for minor fluctuations, while a threshold that is too high prevents the model from adapting to genuine concept shifts. A similar balance is observed for the **Elimination Threshold** ($\tau_e$) in Figure 9(c), where a moderate value is crucial to prune outdated forecasters without prematurely discarding useful ones. Parameters that control adaptation speed, such as the **Local Ratio** ($\tau_l$) in Figure 9(f), also demonstrate a clear optimal range, highlighting the importance of balancing responsiveness to recent data.

Conversely, `CEP` exhibits considerable robustness to other parameters. For instance, the **Gene Ratio** ($\tau_g$) in Figure 9(d) shows stable performance across a wide range of values, indicating that the framework is not overly dependent on the precise weighting between short-term and long-term statistical features. Overall, these findings confirm that while our default settings provide strong baseline performance, targeted tuning of the most sensitive parameters can further enhance results for specific data characteristics.

## C.7 PRACTICAL USAGE GUIDE

While the default hyperparameters provide a robust starting point, performance can be further optimized by tuning them to the specific characteristics of a time series dataset. Based on our analysis, we provide the following practical guidance to help users configure `CEP` for their specific scenarios. Table 11 summarizes key hyperparameters, their effects, and our tuning recommendations.

# D COMPLEXITY ANALYSIS

The efficiency of `CEP` is analyzed in terms of its time and space complexity.

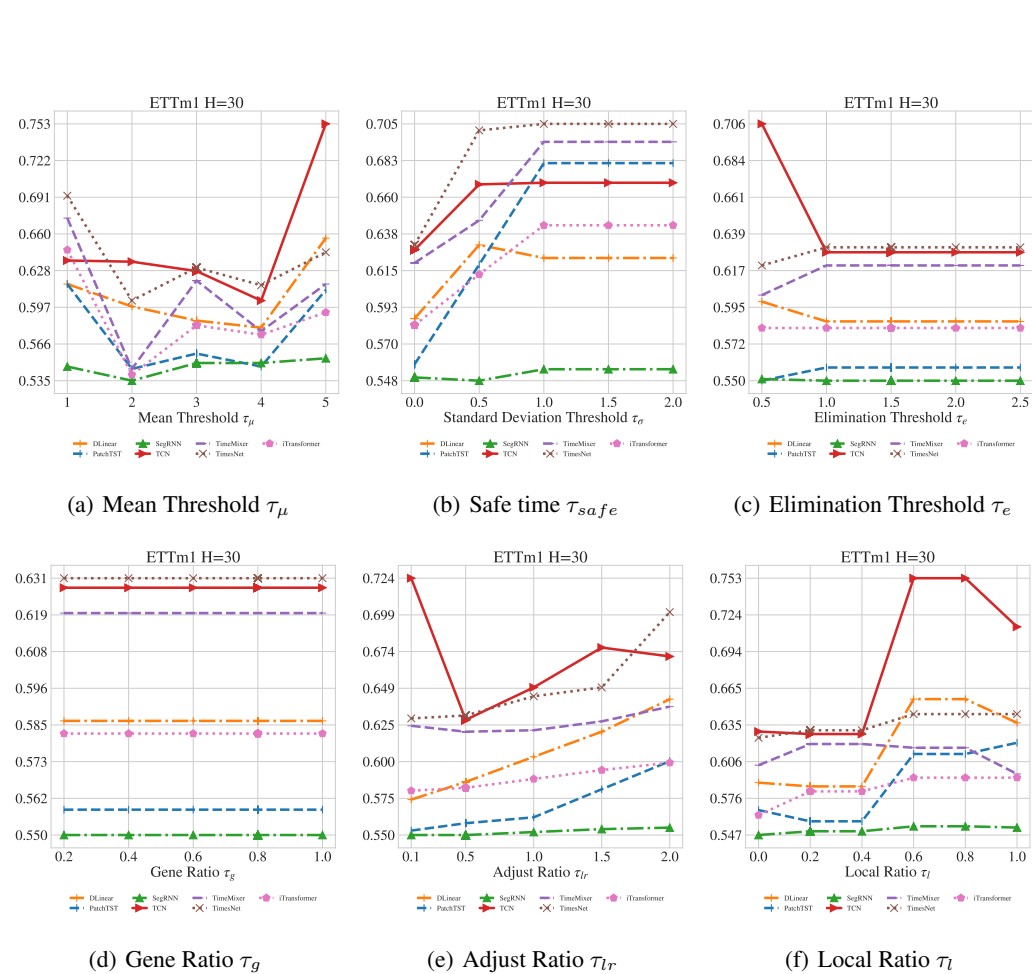

Figure 9: Results of Hyperparameter Sensitivity Analysis

Table 11: Practical guide for tuning `CEP`'s hyperparameters.

| Hyperparameter | Description | Tuning Guidance |
|---|---|---|
| $\tau_\mu$ | *Mean Threshold*: Controls sensitivity to shifts in the data's mean value. | **For noisy data or gradual drifts**, use a higher value (e.g., 3.0-5.0) to prevent creating spurious forecasters. **For abrupt, clear shifts**, a lower value (e.g., 2.0-3.0) enables faster adaptation. |
| $\tau_e$ | *Elimination Threshold*: Determines how long an inactive forecaster is retained. | **For concepts that recur after long intervals** (e.g., annual seasonality), use a higher value to preserve this knowledge. **If memory is a constraint** or concepts are transient, a lower value keeps the pool lean. |
| $\tau_{safe}$ | *Safety Period*: The number of instances a new forecaster must see before it can evolve again. | **For highly volatile or spiky data**, a longer period ensures evolution is based on stable trends, not transient noise. For cleaner data, a shorter period allows for more rapid evolution if needed. |
| $\tau_{lr}$ | *LR Adjustment Ratio*: Sets the initial learning rate multiplier for a newly evolved forecaster. | A small value (e.g., 0.1-0.5) is generally recommended to enforce a sharp LR drop, promoting **stable convergence** on the new concept. Use larger values with caution as they risk unstable training. |
| $\tau_l$ | *Local Gene Ratio*: Controls how quickly the local gene adapts to new instances (EMA decay). | **For rapidly changing environments**, a higher value (e.g., 0.5-0.8) allows for faster tracking. **For smoother trends**, a lower value (e.g., 0.1-0.3) provides more stable local feature representation. |

**Time Complexity.** The time complexity per prediction step is highly efficient. Since `CEP` activates only a single, specialized forecaster from the pool for any given instance, its inference time remains identical to that of the base forecaster model, denoted as $O(k)$. Here, $k$ represents the computational operations required by the base model for a single input.

**Space Complexity.** The space complexity of `CEP` is actively managed to remain practical and efficient. While the theoretical worst-case space complexity could grow linearly with the total number of unique concepts encountered, $O(|\widetilde{\mathcal{D}}| \cdot k)$, this scenario is effectively prevented by the integral *Forecaster Elimination* mechanism. This core component acts as a resource governor, dynamically pruning the pool by removing forecasters that correspond to transient, noisy, or outdated concepts that no longer recur.

Consequently, the actual memory footprint does not scale with the entire history of concepts but rather adapts to the number of *active and relevant* concepts in the current data regime. This design keeps the pool size bounded, ensuring the space complexity is manageable in practice. As empirically demonstrated in Figure 10, the number of active forecasters remains well-controlled throughout the online stage, confirming the effectiveness of this self-regulating mechanism.

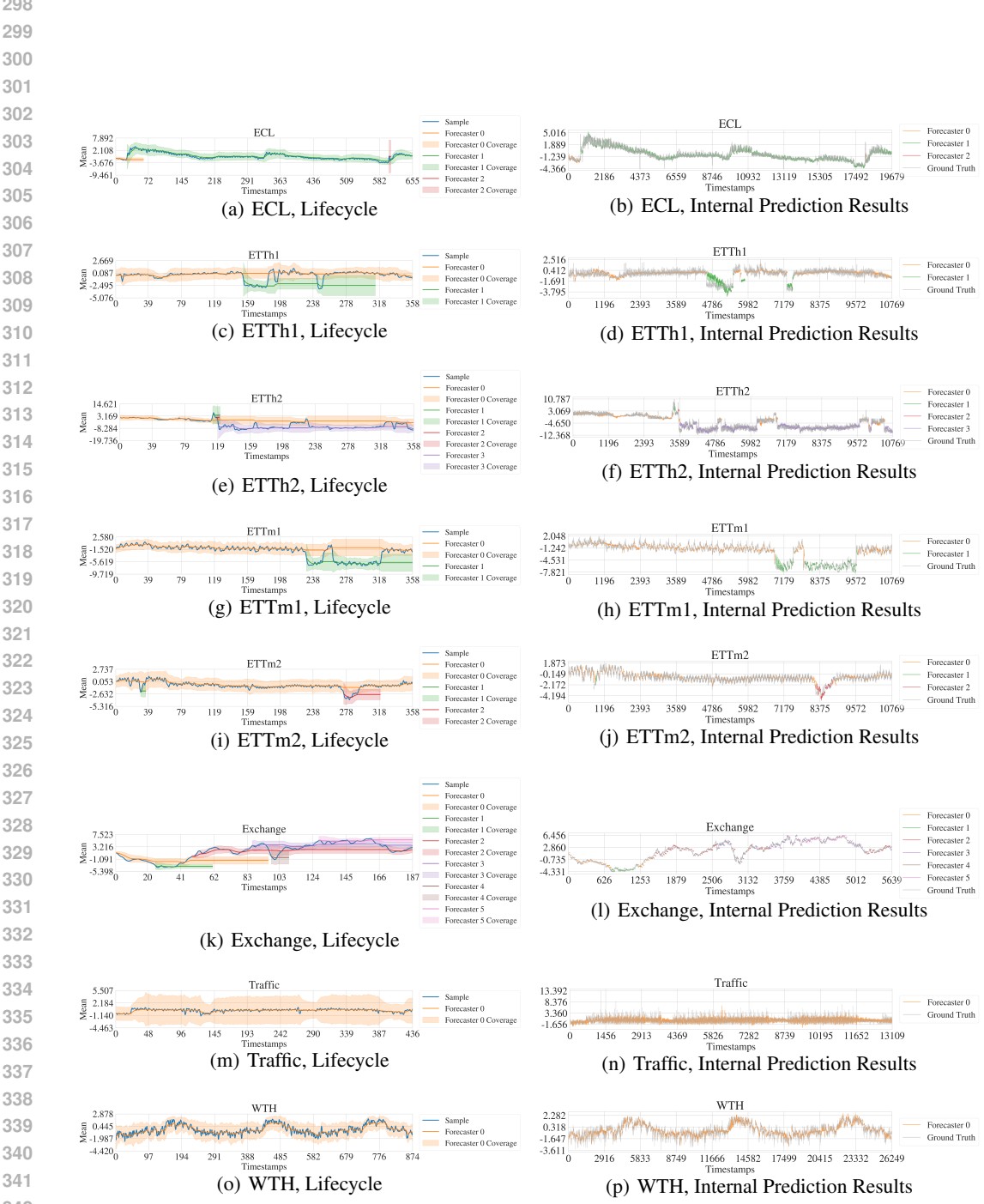

Figure 10: Visualization of forecaster gene trajectories.

