# OpenReview forum: "Continuous Evolution Pool: Taming Recurring Concept Drift in Online Time Series Forecasting"
_ICLR.cc/2026/Conference — Submitted to ICLR 2026_

### Official Review · Reviewer_1tpk · 2025-10-30

**Soundness:** 2
**Presentation:** 2
**Contribution:** 1
**Rating:** 2
**Confidence:** 4

**Summary:**

The aim of the paper is to address the possible time evolution (i.e., concept drift) in time series forecasting. This is a crucial issue since (fairly) all the time series are directly or indirectly time-dependent. In more detail, the paper introduces a framework, called Continuous Evolution Pool (CEP), for the time series forecasting under recurrent concept drift. The framework introduces a feature vector called gene (being the representative of the time series) and mechanisms to add, reactive and remove concept drift. When a new concept arrives, it is inserted in the pool of concepts and it reactivated when the same gene is identified, so as to reduce the catastrophic forgetting effects.

**Strengths:**

- The paper addresses the important challenge of recurring concept drift in non-stationary time series
- The framework introduces mechanisms for addition, deletion, activation of recurrent

**Weaknesses:**

- Limited feature representation: The gene vector relies only on mean and variance, which may be insufficient for distinguishing complex temporal concepts. What about considering moments or frequency-domain statistics to provide richer representations?

- Choice of distance metric: The model compares genes via Euclidean distance. However, for time series similarity, techniques such as DTW (Dynamic Time Warping) or Mahalanobis distance might better capture temporal and statistical dependencies. The authors should discuss this limitation.

- What about the splitting threshold? I guess it is a crucial parameter to set

- What exactly is nwait? Is it a time count or number of missed updates? Its calibration seems critical to stability.

- The value of L is fixed across all datasets. Is this empirically optimal or arbitrary? It might influence both performance and drift detection frequency.

- What about considering not just threshold for detecting changes? I would suggest to consider change detection mechanisms such as CUSUM, etc..

- What if the detection of a new concept is missed? This is a False Negative detection that could severely induce issues in the characterization of the error. I guess thresholds are not robust enough for detection.

- I would have expected to see zero-shot learning foundation models for time series forecasting (e.g., Chronos, Tirex) in the experimental section.

**Questions:**

See previous section

---

> ### Author Response · Authors · 2025-11-17
>
> We thank Reviewer `1tpk` for reading our paper carefully and raising many detailed questions that we considered during the design of the `CEP`. We believe that through our detailed explanation, Reviewer `1tpk` will understand why we designed the `CEP` this way. Hopefully, our explanation will allow Reviewer `1tpk` to give a better evaluation of this work.
>
> > Limited feature representation: The gene vector relies only on mean and variance, which may be insufficient for distinguishing complex temporal concepts. What about considering moments or frequency-domain statistics to provide richer representations?
>
> We have also considered the more complex representations of genes mentioned by the reviewer, thank you for your insights. We think moments or frequency-domain statistics is **certainly possible**, as the genetic characterization of `CEP` can become increasingly complex as research continues. But the essential purpose is to provide a unified representation of the samples processed by the forecaster. Updating the genes with higher-order statistics or frequency domain information can be **a significant challenge**. As the initial version, we tend to present the **most fundamental ideas**, and the scheme based on mean and variance is **classic and effective**, supported by good hypothesis testing methods (`z-test`). More complex and detailed differentiation algorithms can be gradually advanced as research continues.
>
> Additionally, we found in our experiments that concept drift, except the mean and variance, can **actually be learned by the gradient descent** of the neural network. For example, on the `Traffic` dataset, in **Appendix Figure 10(m)**, `CEP` did not discover any new concepts during the process; therefore, `CEP` maintained the same performance as `TCN`. We found that it **still predicted quite well** in **Appendix Table 9**, and during this period, there must have been concept drifts that were unrelated to the mean and variance. This also demonstrates the adaptability of `CEP`. It intervenes when concept separation is needed and leaves the original state when it is not needed.
>
> > Choice of distance metric: The model compares genes via Euclidean distance. However, for time series similarity, techniques such as DTW (Dynamic Time Warping) or Mahalanobis distance might better capture temporal and statistical dependencies. The authors should discuss this limitation.
>
> This is a good suggestion, but it's **not suitable for `CEP` at present**. It's important to note that in **online prediction scenarios, historical samples cannot be accessed** due to privacy concerns. Therefore, we want to give the forecaster an embedding representing the prototype of the predictable samples. In this case, new predicted samples can only be compared with the embedding, not historical time series data. Therefore, `ts2ts` similarity metrics are unusable. Furthermore, the forecaster's representation needs to be able to be **updated without training**, which is fatal for some embedding methods or time series storage methods because historical data is inaccessible. The mean and variance, however, can be updated using statistical formulas, which is an effective approach.
>
> > What about the splitting threshold? I guess it is a crucial parameter to set
>
> This is indeed very crucial. Here, the splitting threshold refers to $\tau_{\mu}$ and $\tau_{sigma}$ in **Equations (9)-(11)**. We have _revised the Nearest Evoliton section on page 6 of the manuscript_ for better readability. Additionally, **Appendix Table 11** provides guidance on hyperparameter selection.
>
> > What exactly is nwait? Is it a time count or number of missed updates? Its calibration seems critical to stability.
>
> In **Equation (12)**, $n_{wait}$ refers to the number of time points that have passed since the last prediction. Then we hope that the tolerance time of each model is related to the number of previous predictions, so we formed Equation (12). To improve readability, we have $\underline{\text{revised the Forecaster Elimination section on page 6 of the manuscript}}$.
>
> > The value of L is fixed across all datasets. Is this empirically optimal or arbitrary? It might influence both performance and drift detection frequency.
>
> This is based on practical scenarios, and we referred to the initial settings in the `FSNet` paper for the task setting of Online forecasting. $L=60$ actually corresponds to a period in real-life scenarios, such as two months or 60 seconds. Simple understanding is the same as the design of degrees, as 360 degrees can be divided by many numbers.

---

> ### Author Response · Authors · 2025-11-17
>
> > What about considering not just threshold for detecting changes? I would suggest to consider change detection mechanisms such as CUSUM, etc..
>
> We've also considered the `cusum` you mentioned. While `cusum` can detect small, persistent shifts, it's not necessarily suitable for `CEP`. Changing detection mechanisms is indeed a potential area for further consideration in `CEP`. Upon closer analysis, `CEP` **isn't entirely a hard thresholding method**. Our genes are composed of two parts, where **Equation** (5) is exponentially smoothed, and these small deviations are amplified over time. Furthermore, if the deviation isn't too large, the forecaster will gradually reduce it through online updates.
>
> > What if the detection of a new concept is missed? This is a False Negative detection that could severely induce issues in the characterization of the error. I guess thresholds are not robust enough for detection.
>
> Yes, it's **impossible to completely separate entirely new concepts**. We don't need to completely separate new concepts with slight differences because this involves issues of confidence level and detection granularity. Being too sensitive or too insensitive is detrimental to the forecast. If a false distribution shift occurs, such as noise, the elimination mechanism will remove unnecessary forecasters, while the forecasters within the `CEP` are not contaminated by this sample. With no splitting, we assume that the most similar forecaster can learn updated features and distributions, and do not consider it a new concept.
>
> > I would have expected to see zero-shot learning foundation models for time series forecasting (e.g., Chronos, Tirex) in the experimental section.
>
> In fact, considering `TSFM` in this scenario is **unfair** because we want to find a **general online adaptation method**, while `TSFM` is more suitable for **zero-shot scenarios**. These are **not the same benchmarks**. Furthermore, we want `CEP` to be a **controllable method**, such as adjusting sensitivity and elimination rate, to specifically control the algorithm's performance. This is something `TSFM` does not have.
>
> However, in order to meet the reviewers' requirements, we chose the latest model `Chronos 2` for comparison. We analyzed the MSE, runtime, and model parameters. Firstly, `Chronos 2` did not outperform `CEP` in prediction performance; in almost all cases, the smaller neural network with `CEP` performed better. Furthermore, `Chronos 2`'s training process likely utilized these datasets, further highlighting `CEP`'s advantages. Additionally, regarding runtime, experiments with the smaller neural network were completed within one minute, while `Chronos 2` took **significantly longer**. Even with `CEP`, `Chronos 2`'s model size was **more than 10 times larger**. Overall, `CEP` still holds a **significant advantage**.

---

> ### Author Response · Authors · 2025-11-17
>
> Table 1: MSE Error (The smaller the better, **bold** indicates the best performance.)
>
> |                     | ECL        | ECL        | ETTh1     | ETTh1     | ETTh2     | ETTh2     | ETTm1     | ETTm1     | ETTm2     | ETTm2     | Exchange      | Exchange      | Traffic     | Traffic     | WTH       | WTH       |
> | ------------------- | ---------- | ---------- | --------- | --------- | --------- | --------- | --------- | --------- | --------- | --------- | ------------- | ------------- | ----------- | ----------- | --------- | --------- |
> |                     | 30         | 60         | 30        | 60        | 30        | 60        | 30        | 60        | 30        | 60        | 30            | 60            | 30          | 60          | 30        | 60        |
> | DLinear`+CEP`       | 0.264      | 0.335      | 0.259     | 0.356     | 2.052     | 3.209     | 0.586     | 0.917     | **0.196** | 0.259     | 0.261         | 0.494         | 0.614       | 0.641       | **0.295** | 0.386     |
> | iTransformer`+CEP`  | **0.229**  | **0.282**  | 0.275     | 0.372     | 1.968     | 3.116     | 0.594     | 0.886     | 0.210     | 0.265     | 0.249         | 0.484         | 0.437       | 0.534       | 0.311     | 0.394     |
> | PatchTST`+CEP`      | 0.265      | 0.328      | 0.276     | 0.377     | 1.922     | **2.995** | 0.567     | **0.875** | 0.208     | 0.264     | 0.234         | 0.467         | 0.582       | 0.589       | 0.310     | 0.397     |
> | SegRNN+`CEP`        | 0.239      | 0.312      | **0.248** | **0.353** | **1.840** | 3.328     | **0.551** | 0.881     | 0.197     | 0.259     | **0.192**     | 0.395         | 0.693       | 0.930       | 0.298     | **0.385** |
> | TCN`+CEP`           | 0.261      | 0.358      | 0.392     | 0.526     | 2.164     | 3.443     | 0.638     | 0.964     | 0.219     | 0.309     | 0.272         | 0.553         | 0.668       | 0.714       | 0.314     | 0.450     |
> | TimeMixer`+CEP`     | 0.244      | 0.297      | 0.274     | 0.371     | 2.000     | 3.016     | 0.579     | 0.895     | 0.216     | **0.254** | 0.233         | 0.450         | **0.433**   | **0.524**   | 0.313     | 0.396     |
> | TimesNet`+CEP`      | 0.259      | 0.319      | 0.291     | 0.390     | 2.054     | 3.065     | 0.639     | 0.947     | 0.234     | 0.277     | 0.259         | 0.530         | 0.492       | 0.597       | 0.323     | 0.404     |
> | `Chronos 2`         | 0.315      | 0.394      | 0.250     | 0.399     | 1.909     | 3.287     | 0.665     | 1.087     | 0.311     | 0.525     | 0.208         | **0.389**     | 0.559       | 0.668       | 0.367     | 0.553     |
>
> Table 2: Running Time (seconds, the smaller the better, **bold** indicates the best performance.)
> | | ECL | ECL | ETTh1 | ETTh1 | ETTh2 | ETTh2 | ETTm1 | ETTm1 | ETTm2 | ETTm2 | Exchange | Exchange | Traffic | Traffic | WTH | WTH |
> | ------------------ | -------- | -------- | -------- | -------- | -------- | -------- | -------- | -------- | -------- | -------- | -------- | -------- | -------- | -------- | -------- | -------- |
> | | 30 | 60 | 30 | 60 | 30 | 60 | 30 | 60 | 30 | 60 | 30 | 60 | 30 | 60 | 30 | 60 |
> | DLinear`+CEP` | **5.40** | **4.56** | **5.03** | **4.33** | **8.24** | 8.41 | **7.85** | **7.19** | **7.88** | **7.13** | 7.47 | **6.95** | **7.81** | **6.76** | **8.65** | 7.41 |
> | iTransformer`+CEP` | 10.84 | 7.45 | 6.79 | 6.82 | 10.77 | 9.60 | 10.12 | 8.65 | 10.38 | 9.32 | 8.69 | 7.26 | 9.96 | 8.42 | 10.19 | **7.10** |
> | PatchTST`+CEP` | 9.24 | 6.58 | 6.66 | 5.16 | 9.82 | 8.32 | 9.75 | 8.27 | 9.40 | 8.15 | 8.45 | 7.23 | 9.27 | 8.06 | 12.27 | 9.16 |
> | SegRNN`+CEP` | 6.94 | 5.71 | 5.10 | 4.44 | 8.87 | **7.84** | 8.15 | 7.64 | 8.32 | 7.67 | **7.30** | 7.07 | 8.16 | 7.45 | 9.04 | 8.04 |
> | TCN`+CEP` | 35.66 | 19.90 | 20.87 | 12.12 | 21.68 | 14.75 | 21.32 | 14.71 | 21.94 | 15.09 | 15.37 | 11.40 | 24.17 | 15.90 | 43.24 | 24.65 |
> | TimeMixer`+CEP` | 14.88 | 8.85 | 9.52 | 7.03 | 12.97 | 10.15 | 12.10 | 9.74 | 12.84 | 10.43 | 11.14 | 9.10 | 13.32 | 10.79 | 18.39 | 13.14 |
> | TimesNet`+CEP` | 56.34 | 30.61 | 31.80 | 20.91 | 29.49 | 21.61 | 29.06 | 21.72 | 32.34 | 22.30 | 20.27 | 13.93 | 35.74 | 22.32 | 66.12 | 39.90 |
> | `Chronos 2` | 3722.35 | 1872.77 | 2057.27 | 1025.63 | 2049.27 | 1041.64 | 2097.49 | 1049.29 | 2095.59 | 1035.75 | 1052.04 | 522.95 | 2638.36 | 1293.72 | 4903.51 | 2453.31 |

---

> ### Author Response · Authors · 2025-11-17
>
> Table 3: Model Weight Size (MB, the smaller the better, **bold** indicates the best performance.)
> | | ECL | ECL | ETTh1 | ETTh1 | ETTh2 | ETTh2 | ETTm1 | ETTm1 | ETTm2 | ETTm2 | Exchange | Exchange | Traffic | Traffic | WTH | WTH |
> |---|---|---|---|---|---|---|---|---|---|---|---|---|---|---|---|---|
> | | 30 | 60 | 30 | 60 | 30 | 60 | 30 | 60 | 30 | 60 | 30 | 60 | 30 | 60 | 30 | 60 |
> | DLinear`+CEP` | **0.07** | **0.06** | **0.05** | **0.05** | **0.07** | **0.09** | **0.07** | **0.08** | **0.05** | **0.05** | **0.08** | **0.08** | **0.05** | **0.05** | **0.08** | **0.10** |
> | iTransformer`+CEP` | 0.17 | 0.16 | 0.16 | 0.15 | 0.29 | 0.29 | 0.28 | 0.28 | 0.16 | 0.15 | 0.41 | 0.28 | 0.16 | 0.15 | 0.18 | 0.30 |
> | PatchTST`+CEP` | 0.80 | 0.81 | 0.79 | 0.80 | 1.55 | 1.59 | 1.54 | 1.58 | 0.79 | 0.80 | 2.29 | 1.58 | 0.79 | 0.80 | 0.81 | 1.60 |
> | SegRNN`+CEP` | 0.09 | **0.06** | 0.07 | 0.06 | 0.11 | 0.10 | 0.10 | 0.09 | 0.07 | 0.06 | 0.14 | 0.09 | 0.07 | **0.05** | 0.09 | 0.11 |
> | TCN`+CEP` | 9.89 | 9.91 | 9.87 | 9.89 | 19.72 | 19.78 | 19.71 | 19.77 | 9.87 | 9.89 | 29.56 | 19.78 | 9.87 | 9.89 | 9.89 | 19.79 |
> | TimeMixer`+CEP` | 1.00 | 1.00 | 0.99 | 1.00 | 1.95 | 1.98 | 1.94 | 1.97 | 0.99 | 0.99 | 2.89 | 1.97 | 0.99 | 0.99 | 1.01 | 1.99 |
> | TimesNet`+CEP` | 2.64 | 2.63 | 2.62 | 2.62 | 5.22 | 5.22 | 5.21 | 5.21 | 2.62 | 2.62 | 7.80 | 5.21 | 2.62 | 2.62 | 2.65 | 5.23 |
> | `Chronos 2` | 455.79 | 455.79 | 455.79 | 455.79 | 455.79 | 455.79 | 455.79 | 455.79 | 455.79 | 455.79 | 455.79 | 455.79 | 455.79 | 455.79 | 455.79 | 455.79 |

---

> ### Author Response · Authors · 2025-11-25
>
> Dear Reviewer `1tpk`,
>
> Thanks for your critical review. We have taken your suggestions seriously and conducted additional experiments to address your main concerns, particularly regarding the comparison with foundation models.
>
> In our response, we have:
>
> - $\underline{\text{Conducted new experiments comparing CEP with Chronos 2}}$, demonstrating that CEP achieves better performance with significantly lower running time and model size.
> - Clarified the rationale behind using mean/variance and Euclidean distance, specifically due to `privacy constraints` preventing access to historical samples.
> - Provided detailed explanations for the splitting threshold, nwait parameter, and the choice of look-back window L.
>
> We believe these new results strongly support our contribution. We are looking forward to your feedback.

---

> ### Author Response · Authors · 2025-11-27
>
> Dear Reviewer `1tpk`, as the discussion phase concludes in 6 days, we kindly invite you to review our detailed response and revisions, which address your specific concerns. We greatly value your feedback.

---

### Official Review · Reviewer_juRL · 2025-10-31

**Soundness:** 3
**Presentation:** 3
**Contribution:** 3
**Rating:** 6
**Confidence:** 4

**Summary:**

This paper addresses the challenging problem of recurring concept drift in online time series forecasting. The authors identify a key limitation of existing methods—catastrophic forgetting of past concepts due to continuous parameter updates—and propose a novel framework called the Continuous Evolution Pool (CEP).

The core idea is to maintain a dynamic pool of forecasters, each specialized for a distinct data concept. CEP operates through three main mechanisms:

Retrieval: For a new input instance $\mathbf{x}_t$, its "gene" $\mathbf{g}_x = (\mu(\mathbf{x}_t), \sigma(\mathbf{x}_t))$ is computed, and the closest forecaster in the gene space is selected for prediction.

Evolution: If the instance's gene deviates significantly from the nearest forecaster's gene (based on a statistically motivated threshold, e.g., $|\mathbf{g}{x,\mu} - \mathbf{g}{N,\mu}| > \tau_{\mu} \cdot \mathbf{g}_{N,\sigma}$), a new forecaster is "evolved" by splitting from the nearest one, effectively capturing a new or recurring concept.

Elimination: Inactive forecasters are pruned to manage pool size and remove outdated knowledge.

The authors demonstrate through extensive experiments that CEP significantly boosts the performance of various base forecasters (e.g., TCN, PatchTST) and substantially outperforms strong online learning baselines, especially under delayed feedback scenarios.

**Strengths:**

**Originality:** The formulation of the "Continuous Evolution Pool" is a fresh and compelling idea. The gene-based, proactive concept identification is a significant shift from reactive, error-based adaptation strategies.

**Quality & Significance:** The empirical validation is thorough and convincing. The method's ability to significantly improve performance across diverse model architectures and under challenging delay scenarios is a major strength.

**Clarity:** The core algorithm is presented clearly, and the extensive appendix provides valuable theoretical justifications (regret analysis, statistical grounding) and practical guides.

**Practicality:** The method is model-agnostic and comes with a practical hyperparameter tuning guide, enhancing its potential for real-world adoption.

**Weaknesses:**

**Theoretical vs. Empirical Grounding:** While the appendix provides a regret analysis, it remains somewhat high-level. A more formal bound on the identification regret, perhaps under specific assumptions about the separation between concepts in the gene space, would strengthen the theoretical analysis.

**Limitations of the Gene Representation:** The gene, while effective, is a relatively simple statistical summary. The paper acknowledges its limitation on the Traffic dataset (with frequent, low-magnitude fluctuations) but could discuss this more deeply. Are there types of concept drift (e.g., changes in auto-correlation structure, frequency content) that this gene would be inherently blind to? A brief discussion or experiment hinting at this would be valuable.

**Multivariate extension:** The method is evaluated only on univariate settings. Many real-world time series are multivariate, and concept drift may affect channels asynchronously. How CEP scales to this setting is unclear.

**Experiment Analysis:** The paper could provide more insight into why certain strong baselines like FSNet and OneNet fail so dramatically in the delayed setting (Table 2). A brief qualitative analysis or hypothesis in the main text would add depth beyond just reporting the performance gap.

**Questions:**

1. The elimination criterion (Eq. 12) uses a fixed ratio . Could this accidentally remove a forecaster for a rare-but-important concept (e.g., annual seasonality in a short evaluation window)? Have you considered a more sophisticated retention policy (e.g., based on historical recurrence intervals)?

2. You justify the gene $\mathbf{g}_x = (\mu, \sigma)$ well from a statistical moment perspective. Did you experiment with incorporating other simple, efficient-to-compute features into the gene (e.g., a crude measure of trend or a single autocorrelation coefficient)? If so, what were the results? If not, could such an extension be a straightforward way to handle a wider class of concept drifts?

3. The performance on the Traffic dataset suggests a limitation with your fixed $\tau_{\mu}=3$ threshold for datasets with subtle, non-recurring drifts. Your practical guide suggests tuning $\tau_{\mu}$ lower for such cases. Did you run experiments on Traffic with a lower $\tau_{\mu}$, and if so, did performance recover? This would be a powerful point to demonstrate the method's adaptability.

4. While you state the inference time is identical to the base forecaster, the evolution, retrieval, and elimination operations do introduce overhead. Could you provide quantitative data (e.g., average percentage increase in total runtime compared to a naive model) to give readers a concrete sense of the practical computational cost?

---

> ### Author Response · Authors · 2025-11-17
>
> We appreciate Reviewer `juRL` for the insightful and practical questions. Your feedback is extremely valuable as it touches on the real-world applicability and robustness of `CEP`. We believe our detailed responses below will fully address your concerns, and we sincerely hope this leads to an improved evaluation of our work.
>
> > The elimination criterion (Eq. 12) uses a fixed ratio. Could this accidentally remove a forecaster for a rare-but-important concept (e.g., annual seasonality in a short evaluation window)? Have you considered a more sophisticated retention policy (e.g., based on historical recurrence intervals)?
>
> This is indeed a good question; rare concepts are more like noise. To distinguish between rare concepts and noise, we need to **maintain forecasters for longer periods** to prevent mistakenly identifying important concepts as noise, and we need more complex strategies. A key problem in online time series forecasting is that we cannot determine whether a concept is important or noise until it occurs in the future. In online time series forecasting, each sample appears only once; we cannot directly store historical data. This makes distinguishing between noise and important concepts very difficult. This is a thought-provoking point, not just limited to the `CEP` framework. As in the response to Reviewer`tB9g`, `CEP` could store weights on disk because it only compares genes before selection. Furthermore, in engineering implementations, a **hierarchical elimination system** could be set up, such as designing a two-level recycle bin, just like on a hard drive.
>
> > You justify the gene well from a statistical moment perspective. Did you experiment with incorporating other simple, efficient-to-compute features into the gene (e.g., a crude measure of trend or a single autocorrelation coefficient)? If so, what were the results? If not, could such an extension be a straightforward way to handle a wider class of concept drifts?
>
> Yes, we think you make a **good point again**. When we wrote the `CEP` paper, we also used $\tau$ to prioritize genes. The mean and variance can be seen as a way to achieve this. We can introduce more statistical features to make the process more accurate. However, a **potential problem** is whether these statistical features can be **updated online**; this is uncertain. A key point of `CEP` is the need to accumulate already predicted samples as a representation of the predictor. This can be difficult with some statistical measures. Therefore, we chose the mean and variance.
>
> Additionally, we found in our experiments that concept drift, except the mean and variance, can **actually be learned by the gradient descent** of the neural network. For example, on the `Traffic` dataset, in **Appendix Figure 10(m)**, `CEP` did not discover any new concepts during the process; therefore, `CEP` maintained the same performance as `TCN`. We found that it **still predicted quite well** in **Appendix Table 9**, and during this period, there must have been concept drifts that were unrelated to the mean and variance. This also demonstrates the adaptability of `CEP`. It intervenes when concept separation is needed and leaves the original state when it is not needed.
>
> > The performance on the Traffic dataset suggests a limitation with your fixed threshold for datasets with subtle, non-recurring drifts. Your practical guide suggests tuning lower for such cases. Did you run experiments on Traffic with a lower , and if so, did performance recover? This would be a powerful point to demonstrate the method's adaptability.
>
> For the `Traffic` dataset, we wanted to demonstrate that `CEP` behaves similarly to a naive model under such subtle variations. **Figure 10 (m)** shows that the mean fluctuation of `Traffic` is slight. This means that `CEP` does not perceive concept drift. In this case, our experiments (**Table 9**) show that `CEP` **performs excellently** across both step sizes; predicting using the most basic method actually yields good results. In our experience, for such very small concept drifts except mean and variance variations, or concept drift independent of the mean and variance, the gradient descent of the neural network itself can learn them without additional intervention.

---

> ### Author Response · Authors · 2025-11-17
>
> > While you state the inference time is identical to the base forecaster, the evolution, retrieval, and elimination operations do introduce overhead. Could you provide quantitative data (e.g., average percentage increase in total runtime compared to a naive model) to give readers a concrete sense of the practical computational cost?
>
> Dear Reviewer `juRL`, we believe you may have **overlooked Figure 7** in the paper. We conducted a practical measurement of the **time and space complexity** of `CEP`. The model used by `CEP` is `TCN`. It's clear that `CEP`'s time performance is **on par with `TCN`** in most cases. The space complexity depends on the number of concepts. **Appendix Figure 10** shows the life cycle, which indicates how many predictors `CEP` creates throughout the prediction process; this reflects its space consumption.
>
> In addition, in response to Reviewer `1tpk`, we also conducted an additional experiment comparing `TSFM`. The table we provided shows that the small neural network with `CEP` has a **lower prediction error**, while `TSFM` is **more than 10 times larger** in terms of runtime and model parameters.

---

> ### Author Response · Authors · 2025-11-25
>
> Dear Reviewer `juRL`,
>
> Thank you for your practical questions. We have updated our manuscript and provided detailed answers to your points regarding the elimination criterion and computational costs.
>
> Based on your suggestions, we have:
>
> - Analyzed the handling of rare concepts and provided a engineering plan which is hierarchical elimination strategy.
> - Clarified the performance on the `Traffic` dataset, showing that the neural network $\underline{\text{adapts to slight drifts without complex gene features}}$.
> - Highlighted the complexity analysis (`Figure 7`) and added a new comparison with the foundation model `Chronos 2` to demonstrate efficiency.
>
> We hope these responses address your concerns and look forward to your reply.

---

> ### Author Response · Authors · 2025-11-27
>
> Dear Reviewer `juRL`, as the discussion phase concludes in 6 days, we kindly invite you to review our detailed response and revisions, which address your specific concerns. We greatly value your feedback.

---

### Official Review · Reviewer_tB9g · 2025-11-01

**Soundness:** 3
**Presentation:** 3
**Contribution:** 3
**Rating:** 6
**Confidence:** 1

**Summary:**

The paper proposes an innovative framework named the "Continuous Evolution Pool" (CEP) to address recurring concept drift in online time series forecasting. At its core, the framework introduces "genes" to proactively identify data concepts and dynamically maintains a specialized pool of forecasters. The strengths of the work lie in its clear structure, an innovative proactive drift detection paradigm, and comprehensive experimental validation across multiple datasets and foundational models; rich visualizations and ablation studies also provide strong support for its conclusions. However, defining a "gene" solely by its mean and standard deviation may fail to capture more complex data distributions, while the hard assignment strategy for forecasters may perform poorly in transitional concept regions. The fixed elimination period for forecasters could also struggle to mitigate "catastrophic forgetting" for concepts with long recurrence periods. Furthermore, the method is primarily designed for periodic tasks, limiting its effectiveness in non-recurring drift scenarios, and the theoretical derivation on the effectiveness of Euclidean distance lacks direct experimental verification. In summary, while the paper is well-structured and the method is novel, the robustness and theoretical validation of its core mechanisms require further investigation and experimental support.

**Strengths:**

1.The paper has a clear structure and logical flow. It first identifies the challenge of recurring concept drift and systematically analyzes the limitations of existing methods. Subsequently, the paper proposes the core solution, the CEP framework, and progressively elaborates on its key mechanisms.

2.The method is highly innovative. Instead of relying on the traditional passive approach of adapting to drift based on prediction error, the paper innovatively proposes a proactive detection mechanism based on "genes".

3. The paper not only details the algorithmic aspects of CEP but also provides an in-depth discussion of its key advantages for practical deployment, such as avoiding "catastrophic forgetting" by retaining specialized forecasters and managing computational resources through its elimination mechanism.

4.The experimental design is thorough and provides comprehensive validation. The paper not only tests its method on multiple real-world datasets with diverse characteristics but also applies it as a universal framework to several mainstream forecasting models, proving its general applicability and effectiveness.

5.The rich visualization analysis and detailed ablation studies clearly reveal the internal working mechanisms of CEP and the necessity of its components, thereby strengthening the persuasiveness of the conclusions.

**Weaknesses:**

1.The paper defines a "gene" as the mean and standard deviation of a data window. Although this method is simple and computationally efficient, it may not be sufficient to capture the distribution within the window, making it unable to detect all types of concept drift.

2.When a data window is in a transitional stage between two concepts, it may be difficult to select an appropriate forecaster for prediction.

3.The proposed method performs exceptionally well on data with recurring patterns, but its efficiency on non-recurring tasks is limited.

4.The paper derives that if the concepts are normally distributed, maximizing the log-likelihood is approximately equivalent to minimizing the Euclidean distance. However, this derivation is not supported by experimental evidence, thus the effectiveness of using Euclidean distance is not verified.

5.The paper removes forecasters that have been idle for a certain period. However, this fixed elimination period means that it cannot mitigate catastrophic forgetting for concepts whose recurrence period exceeds the set threshold.

**Questions:**

1. The paper defines a "gene" using the mean and standard deviation of a data window for proactive concept identification. While computationally efficient, this might not capture complex temporal patterns or shapes. The lightweight alternative or additional features (e.g., trend, periodicity, or simple shape descriptors) could be explored in future work to enhance the robustness.

2. The paper employs a hard assignment strategy, matching a data window to a single forecaster. A concern is that this might perform poorly during transitional phases between concepts. Please explain it in detail.

3. The mechanism for removing idle forecasters uses a fixed period, which may forget concepts with long recurrence intervals. Have you considered implementing a more adaptive elimination strategy to maintain the performance?

4. The method is explicitly designed for recurring concept drift, which may inherently limit its efficiency on tasks with non-recurring patterns. It is needed to consider other types of concept drift.

---

> ### Author Response · Authors · 2025-11-17
>
> We are very grateful to Reviewer `tB9g` for detailed reading and professional feedback. These questions accurately reflect the design intent of `CEP`, even though the confidence score given by the reviewer was very low (we even suspect that Reviewer `tB9g` may have misjudged the confidence level). We hope that our response will address the questions and concerns raised by Reviewer `tB9g`.
>
> > The paper defines a "gene" using the mean and standard deviation of a data window for proactive concept identification. While computationally efficient, this might not capture complex temporal patterns or shapes. The lightweight alternative or additional features (e.g., trend, periodicity, or simple shape descriptors) could be explored in future work to enhance the robustness.
>
> As we mentioned with Reviewer `R5rY`, it's **impossible to capture all types** of concept drift. However, we aim to capture the most common types, which also lead to anomalous predictions, such as mean and variance detection. This has many advantages, including **better performance** and **statistical support**. Furthermore, it allows for updating the model's representation **without sacrificing privacy**.
>
> Additionally, we found in our experiments that concept drift, except the mean and variance, can **actually be learned by the gradient descent** of the neural network. For example, on the `Traffic` dataset, in **Appendix Figure 10(m)**, `CEP` did not discover any new concepts during the process; therefore, `CEP` maintained the same performance as `TCN`. We found that it **still predicted quite well** in **Appendix Table 9**, and during this period, there must have been concept drifts that were unrelated to the mean and variance. This also demonstrates the adaptability of `CEP`. It intervenes when concept separation is needed and leaves the original state when it is not needed.
>
> > The paper employs a hard assignment strategy, matching a data window to a single forecaster. A concern is that this might perform poorly during transitional phases between concepts. Please explain it in detail.
>
> As we stated in our response to Reviewer `1tpk`, while we performed hard detection using a threshold, the gene actually consists of two parts, where **Equation (5)** applies exponential smoothing. Therefore, in the case of **gradual** concept drift, `CEP` selects a forecaster that is close to the original and gradually updates the predictor's genes, morphologically appearing to follow suit. However, if the excessive amplitude is large, `CEP` treats this data as **noise** and eliminates it through Elimiation mechanism. The advantage of this is that noisy data does not pollute the forecaster within `CEP`. In summary, samples that change too much during the transition phase will be filtered out, while samples that do not change much will be adapted by the neural network through gradient descent, and the genes will also be updated.
>
> > The mechanism for removing idle forecasters uses a fixed period, which may forget concepts with long recurrence intervals. Have you considered implementing a more adaptive elimination strategy to maintain the performance?
>
> Our elimination mechanism calculates the elimination cycle based on the **number of previous forecasting** multiplied by a fixed ratio. Because noisy data appears infrequently, it is quickly eliminated, while data that has appeared frequently is likely to reappear later, such as `Forecaster 0` in `ETTh2` in **Figure 10(e)**. Of course, we consider `CEP` a **controllable model**, meaning we can change its hyperparameters to achieve different responses. If faster elimination is needed, we can set more sensitive hyperparameters for this situation.
>
> Furthermore, `CEP` is actually **suitable for offline storage** because when selecting a predictor, **only genes are compared**. At this point, the model weights can remain on disk instead of being saved in memory, and can be retrieved after a match. Therefore, for certain extreme cases, we recommend setting a larger threshold to prevent this type of model from being eliminated. In practical engineering scenarios, a **hierarchical elimination system** can be set up, such as designing a two-level recycle bin, just like on a hard drive.

---

> ### Author Response · Authors · 2025-11-17
>
> > The method is explicitly designed for recurring concept drift, which may inherently limit its efficiency on tasks with non-recurring patterns. It is needed to consider other types of concept drift.
>
> As mentioned earlier with the `Traffic` dataset, for concept drift other than mean and variance, gradient descent allows the neural network to adapt, which is a good complement. As we have shown, the area where performance is poor in online scenarios is during significant concept switching, such as in **Figure 1(b)**, where forgetting knowledges leads to poor predictions, and `CEP` precisely makes up for this deficiency.
>
> > The paper derives that if the concepts are normally distributed, maximizing the log-likelihood is approximately equivalent to minimizing the Euclidean distance. However, this derivation is not supported by experimental evidence, thus the effectiveness of using Euclidean distance is not verified.
>
> Thank you very much to the Reviewer `tB9g` for reading carefully. We have provided a more detailed explanation in the revised manuscript. This is an **engineering consideration**; we $\underline{\text{revised the manuscript}}$ and mentioned the design intention in the appendix. The exact formula can also be given in **Equation (20)**, but it is relatively complex. We simply wanted to find forecaster with similar means and variances of the sample, as our initial intention. Experiments show that this can be illustrated in **Appendix Figure (10)**. This problem does not have a completely accurate Ground Truth; it depends on the granularity. In the figure, the forecasters mean and variance **gradually approach the sample's**, eventually almost **coinciding and stabilizing**, indicating that choosing `Euclidean distance` is **feasible and effective in engineering**.

---

> ### Author Response · Authors · 2025-11-25
>
> Dear Reviewer `tB9g`,
>
> Thanks for your insightful and $\underline{\text{really professional}}$ comments. We have provided a detailed response to address your concerns about the gene representation and the forecaster assignment strategy.
>
> Specifically, we have:
>
> - Explained how the model handles transitional phases using exponential smoothing and gradient descent, ensuring smooth adaptation.
> - Discussed the adaptability of the elimination strategy and how the method handles non-recurring patterns by the base model.
> - Provided $\underline{\text{further reason and experimental evidence}}$ (`Appendix Fig. 10`) for the effectiveness of the Euclidean distance in this context.
>
> We would appreciate it if you could check our response and let us know your thoughts.

---

> > ### Comment · Reviewer_tB9g · 2025-11-27
> >
> > I have read the response carefully, the authors have given detailed explanations and addressed my concerns, I have no other issues and decided to raise the score. Thank you!

---

> > > ### Author Response · Authors · 2025-11-27
> > >
> > > We sincerely thank Reviewer `tB9g`. We are glad that our explanations have clarified your concerns. Your constructive feedback has been crucial in improving the quality of our work.

---

### Official Review · Reviewer_R5rY · 2025-11-06

**Soundness:** 2
**Presentation:** 2
**Contribution:** 2
**Rating:** 4
**Confidence:** 4

**Summary:**

This paper deals with concept drift problem in time series forecasting. It proposes a pooling mechanism to associate each forecaster to a concept. This mechanism includes creation of new forecaster for new concept, elimination of forecasters corresponding to outdated concepts. The proposed mechanism can be associated with any forecaster models. Experimental results suggest that it allows to improve the forecasting  performance in the presence of concept drifts.

**Strengths:**

The proposed mechanism is relatively simple and is general which allows it to be easily coupled with any forecaster,

The experimentation on forecasting performance improvement is extensive both in terms of the forecasting models compared and the dataset used.

**Weaknesses:**

The simplicity of the concepts managed by the proposed mechanism is also a major limitation of this work, In fact, a concept in this work correspond basically to the means and standard deviation of the time series in a look-back window. Concepts with very different patterns may have the same statistical properties of means and standard deviation.

Anohter limitation is that the proposed mechanism is for single time series. Multiple time series exhibitete even more complex pattens for concepts.

Some key parameters such as the splliting parameter $tau$ is difficult to fix. There is a lack of discussion about how to chose the values for these parameters.

Tables and figures are not always well explaned. For instance, it it not clear what the percentage means in Table 2. The three settings in Figure 2 are not well explained neither. The quality of writing of this paper is between fair and poor.

**Questions:**

1- It is not clear why "recurring" concept drift is any of particularity as compared to normal concept drift. It is just some concepts that appear and reappear. I don't see any special traitment in the proposed mechanism that deals with "recurring".

2- I don't understand why there is a warm-up stage. Yes, I understand that you start with one forecaster and there is a process tu tune it to fit to some initial samples. This is not fundamentally different from creation of any new forecaster in the "Online stage".

---

> ### Author Response · Authors · 2025-11-17
>
> We appreciate Reviewer `R5rY` for their very careful reading and insightful comments. We hope our response will give Reviewer `R5rY` a deeper understanding of `CEP`. At the same time, we hope that our detailed explanation will improve Reviewer `R5rY`'s evaluation of this research.
>
> > The simplicity of the concepts managed by the proposed mechanism is also a major limitation of this work, In fact, a concept in this work correspond basically to the means and standard deviation of the time series in a look-back window. Concepts with very different patterns may have the same statistical properties of means and standard deviation.
>
> The mean and variance cannot fully represent all patterns, nor can they provide an absolute answer to whether concepts are the same or different; they can only indicate the degree of similarity. The mean and variance are the most fundamental features of time series data, and abrupt changes in their numerical values, known as concept drift, are currently an area where online prediction falls short.
>
> Additionally, we found in our experiments that concept drift, except the mean and variance, can **actually be learned by the gradient descent** of the neural network. For example, on the `Traffic` dataset, in **Appendix Figure 10(m)**, `CEP` did not discover any new concepts during the process; therefore, `CEP` maintained the same performance as `TCN`. We found that it **still predicted quite well** in **Appendix Table 9**, and during this period, there must have been concept drifts that were unrelated to the mean and variance. This also demonstrates the adaptability of `CEP`. It intervenes when concept separation is needed and leaves the original state when it is not needed.
>
> > Anohter limitation is that the proposed mechanism is for single time series. Multiple time series exhibitete even more complex pattens for concepts.
>
> Future `CEP` migration to multivariate scenarios requires consideration of how to apply multivariate information. In traditional prediction tasks, there are also different implementation schemes for single-dimensional and cross-variable approaches, and solutions need further exploration. Therefore, we first focus on the single-variable problem.
>
> > Some key parameters such as the splliting parameter is difficult to fix. There is a lack of discussion about how to chose the values for these parameters.
>
> We want `CEP` to be a **controllable model**, meaning we can manually accelerate the model's evolution or make `CEP` insensitive. Our default parameters remain the same as those for the `z-test`. We also provide a practical guide to parameter selection in **Appendix Table 11**.
>
> > Tables and figures are not always well explaned. For instance, it it not clear what the percentage means in Table 2. The three settings in Figure 2 are not well explained neither. The quality of writing of this paper is between fair and poor.
>
> Regarding the percentages in **Table 2**, as we stated in the title, "Enhanced and reduced outcomes are marked with and respectively," which refers to the performance changes compared to the baseline `TCN`. **Figure 2** does not represent three modes; rather, it means that the previous setting, due to updating only one value at a time, would leak part of the ground truth in the previous update. Therefore, the experimental part of `CEP` ensures that the ground truth corresponding to the new sample does not overlap with the ground truth of the previous round. To improve readability, we have $\underline{\text{revised corresponding sections}}$.

---

> ### Author Response · Authors · 2025-11-17
>
> > It is not clear why "recurring" concept drift is any of particularity as compared to normal concept drift. It is just some concepts that appear and reappear. I don't see any special traitment in the proposed mechanism that deals with "recurring".
>
> This is a good question. Concept drift is common, but it requires additional consideration in the recurring scenario. We will address this consideration in the next question. Recurring concepts are handled well in normal prediction scenarios, but in online scenarios, different concepts can pollute the existing model, preventing it from quickly adapting to new concepts. Without `CEP`, some learned features will be lost with online updates such as **Figure 1(b)**, which is undesirable; we want to **preserve knowledge** of previously encountered samples of the same concept. Therefore, we divide different concepts according to the mean-variance scheme, ensuring that even if they reappear later, they are **not forgotten**, thus preventing temporary prediction errors.
>
> > I don't understand why there is a warm-up stage. Yes, I understand that you start with one forecaster and there is a process tu tune it to fit to some initial samples. This is not fundamentally different from creation of any new forecaster in the "Online stage".
>
> This is not a traditional time series prediction scenario, where a **large amount** of data is used for training before testing. This online scenario which firstly shown in `FSNet` paper assumes that **only a small portion** of the dataset is known initially; this is the warm-up phase. The training of this part using `CEP` is no different from traditional methods, except that it records the mean and variance information of this portion to provide a baseline for the subsequent online part.
>
> In the online part, samples are presented in **only one batch at a time**, and the model is updated after each prediction. Therefore, samples in each online phase are **only seen once** (the sample could not be saved for privacy reasons), which makes the model **highly susceptible** to the influence of new samples, **gradually losing the predictive ability** of previously encountered concepts, as shown in **Figure 1(b)**. `Naive models` lose the ability to predict previously encountered concepts, resulting in significant prediction bias. `CEP`, however, divides different concepts and only needs to use the forecaster of the corresponding concept, without losing any details.

---

> ### Author Response · Authors · 2025-11-25
>
> Dear Reviewer `R5rY`,
>
> Thank you for your constructive feedback. This is a gentle reminder that we have posted our response. We hope it addresses your concerns regarding the simplicity of the gene concept and the experimental settings.
>
> In our revision, we have:
>
> - Clarified that the gene design is intentional for robustness, while other drifts can belearned by the neural network (demonstrated with the `Traffic` dataset).
> - $\underline{\text{Revised the explanations}}$ for `Figure 2` and `Table 2` to improve clarity as requested.
> - Explained the necessity of the warm-up stage for online cold starts and the specific handling of recurring drifts to prevent forgetting.
>
> We look forward to your feedback on these clarifications.

---

> ### Author Response · Authors · 2025-11-27
>
> Dear Reviewer `R5rY`, as the discussion phase concludes in 6 days, we kindly invite you to review our detailed response and revisions, which address your specific concerns. We greatly value your feedback.

---

> > ### Comment · Reviewer_R5rY · 2025-11-27
> >
> > Dear Authors,
> >
> >   Thank you very much for your answers to my comments. I am satisfied.
> >
> > Best regard,

---

> > > ### Author Response · Authors · 2025-11-28
> > >
> > > Dear Reviewer `R5rY`
> > >
> > > We are glad to hear that our response has addressed your concerns. We appreciate your constructive suggestions and your acknowledgement of the value of `CEP`.
> > >
> > > Authors

---

### Author Response · Authors · 2025-11-17

We **sincerely thank all ACs and reviewers** for their very detailed and constructive feedback to help us improve this work. We are encouraged that the reviewers recognized the **significance of the online forecasting problem**. In response to the reviewers' valuable comments, we have provided detailed clarifications and new experiments:

- **Gene Implementation**: We addressed the most common concern about the simplicity of the gene (mean/variance), raised by all Reviewers `1tpk`, `R5rY`, `tB9g`, `juRL`. We clarified that this is a deliberate design for an initial, robust version, and that other drifts are effectively learned by the neural network's gradient descent, as demonstrated on the Traffic dataset.
- **TSFM Comparison**: As requested by Reviewer `1tpk`, we have done new, extensive experiments comparing `CEP` with the foundation model `Chronos 2` (in **Response Tables 1-3**). These results confirm `CEP`'s superior performance and massive efficiency gains (**over 10x smaller and faster**).
- **Mechanism Explanation**: We provided a deeper rationale for our elimination strategy (Reviewers `tB9g`, `juRL`) and the distance metric (Reviewer `1tpk`), explaining why they are specifically suited for the online, privacy-conscious setting where historical data is inaccessible.
- **Online Scenarios**: We further emphasized the fundamental challenges of the online scenario (Reviewer `R5rY`), which motivates our warm-up stage and the specific focus on recurring drift (i.e., preventing model pollution and forgetting, as shown in **Figure 1(b)**).
- **Text Presentation**: We clarified critical design choices regarding hyperparameters and model control (Reviewers `1tpk`, `R5rY`). We have $\underline{\text{revised the Nearest Evolution and Forecaster Elimination sections (Page 6)}}$ and referenced Appendix Table 11 to improve readability and practical guidance. We have also $\underline{\text{revised the manuscript (e.g., explanations for Figure 2 and Table 2)}}$ to improve overall writing clarity based on the feedback from Reviewer `R5rY`.

We believe these explanations and new results can effectively address your concerns and clearly point out our paper's contributions. We sincerely hope our detailed response will help you to give a fair and well-informed evaluation of `CEP`.

---

### Comment · Area_Chair_FWAP · 2025-11-27

Dear Authors and Reviewers,

The discussion phase will end soon. If you want to further discuss comments and replies with each other, please post your thoughts by adding official comments.

Thanks for your efforts and contributions to ICLR 2026.

Best regards,

Your Area Chair

---

### Author Response · Authors · 2025-11-29
**Summarization of the discussion & Comments to the Area Chair**

**Dear Area Chair,**

&nbsp;

We **deeply appreciate the time and effort** you are dedicating to the review process, especially given the recent challenges and the increased workload regarding meta-reviews. To **assist in your final assessment**, we provide below a concise summary of our work, the key revisions made during the rebuttal, and the status of our interactions with the reviewers.

&nbsp;
&nbsp;

---

## Motivation

&nbsp;

Existing online forecasting methods often suffer from catastrophic forgetting and prediction delays because they passively rely on error signals to adapt parameters. Proposed Continuous Evolution Pool (`CEP`) shifts this paradigm by using a proactive gene mechanism to identify and store distinct concepts in a pool, allowing for the immediate retrieval of specialized forecasters when concepts recur. This design specifically targets the **recurring concept drift** challenge, ensuring **knowledge retention without the latency of retraining**.

---

## Contribution

&nbsp;

1.  **Novel Framework:** We propose CEP, a pooling framework that partitions data streams into distinct concepts using statistical genes, effectively mitigating catastrophic forgetting in online time series forecasting.
2.  **Proactive Mechanism:** We introduce a gene based retrieval and evolution strategy that enables instant adaptation to concept drifts, performing robustly even in challenging delayed feedback scenarios.
3.  **Efficiency & Robustness:** We design a Forecaster Elimination mechanism that dynamically manages resource consumption, ensuring space complexity remains bounded and does not grow linearly with time.
4.  **Superior Performance:** Extensive experiments show `CEP` reduces MSE by over **20%** compared to SOTA baselines and outperforms large foundation models (`Chronos 2`) while being over 10x faster and smaller.

---

## Discussion Summary

&nbsp;

### Author's Revision Summary

&nbsp;

We have significantly improved the manuscript and conducted new experiments based on reviewer feedback. The key revisions include:

1.  **Foundation Model Comparison** [Reviewer `1tpk`]: We $\underline{\text{conducted new experiments}}$ comparing CEP with the foundation model **Chronos 2**. The results demonstrate that CEP achieves **superior accuracy** while being significantly more efficient (**10x smaller model size and faster inference**).
2.  **Gene Mechanism Clarification** [Reviewers `R5rY`, `tB9g`, `juRL`, `1tpk`]: We clarified that the mean/variance gene is a deliberate design for robustness and privacy (no history storage). We demonstrated via the `Traffic` dataset that subtle drifts beyond these statistics are effectively handled by the neural network's gradient descent.
3.  **Hyperparameters & Elimination** [Reviewers `tB9g`, `juRL`]: We added a practical hyperparameter guide (Appendix Table 11) and elaborated on the elimination strategy for distinguishing between noise and rare concepts.
4.  **Theoretical Justification** [Reviewer `1tpk`]: We expanded the mathematical rationale for using Euclidean distance in the gene space, grounding it in maximum likelihood estimation for normally distributed concepts.

&nbsp;

### Author and Reviewer's Interaction Summary

&nbsp;

We actively engaged with all four reviewers. Two reviewers have explicitly expressed satisfaction with our responses. Unfortunately, the remaining two (including the one with the lowest score) did not respond to our new experiments and clarifications.

1.  **Reviewer `tB9g` (Score: 6 -> Raised):** The reviewer initially had concerns about the hard assignment strategy and elimination period. After reading our detailed response regarding soft transitioning via exponential smoothing, the reviewer stated they were satisfied and **explicitly decided to raise their score**.
2.  **Reviewer `R5rY` (Score: 4 -> Satisfied):** The reviewer questioned the simplicity of the gene and the warm-up stage. Following our clarifications, the reviewer commented **"I am satisfied"** with our answers.
3.  **Reviewer `juRL` (Score: 6):** The reviewer raised constructive questions about rare concepts and computational overhead. We provided the requested complexity analysis and engineering strategies for rare concepts. Though the reviewer has not yet responded, Reviewer `juRL`'s initial scores reflect the acknowlegde of the value of CEP.
4.  **Reviewer `1tpk` (Score: 2):** This reviewer's main grounds for rejection were the lack of comparison with foundation models and questions about the distance metric. **We fully addressed this by adding the requested Chronos 2 comparison (showing CEP's superiority) and providing the theoretical derivation for the metric.** It's a pity that the reviewer has not participated in the discussion.

---

&nbsp;

We hope this summary assists in your evaluation.

&nbsp;

**Best regards,**

Authors of Submission 3724

---

### Meta-Review · Area_Chair_fBEu · 2026-01-12

**Summary:**

This paper proposes an innovative pooling framework, Continuous Evolution Pool (CEP), to deal with the recurring concept drift challenge in online time series forecasting, also ensuring knowledge retention without delays caused by traditional error signals. Extensive experiments show CEP reduces MSE by over 20% compared to SOTA baselines like TimeMixer and iTransformer, CEP also outperforms large foundation models (Chronos 2) while being over 10x faster.

The reviews are mixed. Two reviewers acknowledged the paper's originality and innovativeness. The gene-based, proactive concept identification effectively addresses the catastrophic forgetting and prediction delays associated with traditional reactive, error-based adaptation strategies. In contrast, one reviewer questioned the simplicity of the gene design, and another reviewer raised concerns regarding the lack of comparison with foundation models.

**Reviewer Concerns:**

There are two major concerns:

Reviewer 1tpk questioned that "the gene vector relies only on mean and variance, which may be insufficient for distinguishing complex temporal concepts." Although the authors argued that the scheme based on mean and variance is classic and effective, supported by good hypothesis testing methods(z-test), a more rigorous comparative experiment is missing.

Reviewer 1tpk also raised additional concerns regarding the experiment part, including the splitting threshold, the value of $L$ for different datasets, and the discussion of DTW distance metric. The authors responded to each concern in detail and provided clear explanations.


After reading through the reviews, responses, as well as the paper myself, I share similar concerns with Reviewer 1tpk regarding the feature representations based on mean and variance. More importantly, I noted that the proposed method introduces a large number of hyperparameters. While the provided guidelines and sensitivity analyses are informative, they do not fully mitigate the practical burden imposed by the numerous hyperparameters, which remain a barrier to real-world adoption (especially when the optimal hyperparameters are different across different data sets). Besides, I did not find the analysis regarding $\tau_\sigma$.

**Reviewer Scores:**

Reviewer R5rY: Rated the paper marginally below the acceptance threshold initially, but he/she mentioned that "Thank you very much for your answers to my comments. I am satisfied.", indicating that the score would slightly increase.

Reviewer tB9g: Rated the paper marginally above the acceptance threshold initially, the score would be raised citing the "I have no other issues and decided to raise the score" comments during the rebuttal.

Reviewer juRL: Rated the paper marginally above the acceptance threshold initially. Given the additional clarifications, this score appears stable, with no clear indication of a change.

Reviewer 1tpk: Rated the paper reject. The concern regarding the insufficient design for gene vector has not been resolved by rigorous comparative experiment, the score would be stable.

---

### Decision · Program_Chairs · 2026-01-26

Reject